# Genetic and Environmental Factors Co-Contributing to Behavioral Abnormalities in *adnp*/*adnp2* Mutant Zebrafish

**DOI:** 10.3390/ijms25179469

**Published:** 2024-08-30

**Authors:** Yongxin Wang, Xiaoyun Sun, Bo Xiong, Ming Duan, Yuhua Sun

**Affiliations:** 1Key Laboratory of Breeding Biotechnology and Sustainable Aquaculture, Institute of Hydrobiology, Chinese Academy of Sciences, Wuhan 430072, China; wangyongxin@ihb.ac.cn; 2University of Chinese Academy of Sciences, Beijing 100049, China; sunnyxyun@usc.edu.cn; 3Department of Forensic Medicine, Tongji Medical College, Huazhong University of Science and Technology, Wuhan 430030, China; bxiong@hust.edu.cn; 4The Innovation of Seed Design, Chinese Academy of Sciences, Wuhan 430072, China; 5Hubei Hongshan Laboratory, Wuhan 430070, China

**Keywords:** *adnp*, *adnp2*, environmental endocrine disruptors, neurodevelopmental disorders, zebrafish

## Abstract

Human mutations of *ADNP* and *ADNP2* are known to be associated with neural developmental disorders (NDDs), including autism spectrum disorders (ASDs) and schizophrenia (SZ). However, the underlying mechanisms remain elusive. In this study, using CRISPR/Cas9 gene editing technology, we generated *adnp* and *adnp2* mutant zebrafish models, which exhibited developmental delays, brain deficits, and core behavioral features of NDDs. RNA sequencing analysis of *adnpa^−^^/^^−^*; *adnpb^−/^^−^* and *adnp2a^−/^^−^*; *adnp2b^−/^^−^* larval brains revealed altered gene expression profiles affecting synaptic transmission, autophagy, apoptosis, microtubule dynamics, hormone signaling, and circadian rhythm regulation. Validation using whole-mount in situ hybridization (WISH) and real-time quantitative PCR (qRT-PCR) corroborated these findings, supporting the RNA-seq results. Additionally, loss of *adnp* and *adnp2* resulted in significant downregulation of pan-neuronal HuC and neuronal fiber network α-Tubulin signals. Importantly, prolonged low-dose exposure to environmental endocrine disruptors (EEDs) aggravated behavioral abnormalities in *adnp* and *adnp2* mutants. This comprehensive approach enhances our understanding of the complex interplay between genetic mutations and environmental factors in NDDs. Our findings provide novel insights and experimental foundations into the roles of *adnp* and *adnp2* in neurodevelopment and behavioral regulation, offering a framework for future preclinical drug screening aimed at elucidating the pathogenesis of NDDs and related conditions.

## 1. Introduction

Neurodevelopmental disorders (NDDs) refer to a heterogeneous group of nervous system disorders caused by changes in brain development, including autism spectrum disorder (ASD), attention deficit/hyperactivity disorder (ADHD), intellectual disability (ID), schizophrenia (SZ), and microcephaly [1,2,3,4]. Genome-wide association studies have identified numerous genes implicated in NDDs, with *ADNP* and *ADNP2* emerging as prominent candidates [5,6,7,8]. *ADNP* is notably associated with ASD and developmental delay (DD), while the *ADNP2* gene is closely linked with SZ, post-traumatic stress disorder (PTSD), ASD, and DD [9,10,11,12,13,14].

ADNP is a vasoactive intestinal peptide (VIP) response gene, which regulates autophagy by inhibiting P53 expression, providing cell protection and ensuring brain structure [15,16,17]. It contains a neuroprotective peptide (NAP, NAPVSIPQ, davunetide) that interacts with neuronal microtubules, which is essential for neuronal survival and function [18]. ADNP2, which is highly expressed in neuronal/glial-like cells, is poorly studied [10]. ADNP and ADNP2 both have zinc finger and homeobox domain, suggesting that they may perform similar or identical functions within cells [19]. Our laboratory’s previous research showed that ADNP influences neuronal differentiation by regulating the Wnt/β-catenin signaling pathway [8], but the specific role of ADNP2 in the nervous system remains unclear. Although mouse models of ASD mimicking *Adnp* mutations have been established and have helped elucidate the etiology of related diseases, the role of *ADNP* in the nervous system and how to treat diseases caused by *ADNP* deletion are still not fully understood [20,21]. Moreover, there are no reports on the development of *Adnp2* knockout lines. Zebrafish have been widely used for modeling NDDs [22,23,24]. In this study, we aim to comprehensively understand and investigate the intrinsic regulatory mechanisms of ADNP and ADNP2 on nervous system function by constructing *adnp*/*adnp2* mutant zebrafish models.

The prevalence of NDDs has increased significantly over the past several decades. Environmental pollutants can potentially increase the risk of NDDs or accelerate its progression [25,26,27]. There is growing evidence suggesting that gene–environment interactions contribute to a large portion of the phenotypic variation in NDDs [28]. Bisphenol A (BPA, 2,2-bis (40-hydroxyphenyl) propane) is a ubiquitous chemical used in the synthesis of polycarbonate plastic and epoxy resins [29,30], while perfluorooctane sulfonic acid (PFOS) is one of the most dominate perfluoroalkyl and polyfluoroalkyl substances (PFASs) detected in the soil and water worldwide [31,32]. Both BPA and PFOS are released into the environment as byproducts of incomplete combustion of industrial wastes [33]. They are known as environmental endocrine disruptors (EEDs), and, importantly, are risk factors for NDDs. The exact roles of the BPA and PFOS in NDDs remain elusive. However, BPA is thought to exert its effects by acting as an agonist of estrogen receptor (ER), or antagonist of androgen receptor (AR) [34], and PFOS has been shown to exhibit concentration-dependent antagonistic effects on human GABA receptor, which is important for early brain development and cortical plasticity [35,36].

This study established zebrafish models of NDDs by deleting the *adnp* gene family, thoroughly analyzed their phenotypic and behavioral characteristics, and elucidated the underlying mechanisms of these abnormal behaviors, providing new experimental foundations for understanding the role of the *ADNP* gene family in NDDs. Additionally, we found that EEDs exacerbated behavioral abnormalities in these mutant larvae, deepening our understanding of the complex interaction between genetic mutations and environmental factors in neurodevelopmental disorders.

## 2. Results

### 2.1. Generation of andp2a^−/−^; 2b^+/−^ Zebrafish

In zebrafish genome, the *adnp* gene has two copies, *adnpa* and *adnpb* [8]. An *adnpa*^−^*^/^*^−^; *adnpb*^−^*^/^*^−^ line was established and maintained in our lab [8]. Additionally, zebrafish *adnp2* also has two paralogues: *adnp2a* and *adnp2b* [19]. To elucidate the function of Adnp and Adnp2, we generated *adnp2a* and *adnp2b* mutant zebrafish by using the CRISPR/Cas9 technology. The DNA sequencing results confirmed a 13-base pair insertion in exon 2 of the *adnp2a* gene and an 11-base pair deletion in exon 4 of the *adnp2b* gene. (Figure 1A).

Maternal zygotic *adnp2a* mutant embryos appeared normal and developed into fertile adults. Conversely, when *adnp2b* heterozygous were crossed, we found that *adnp2b* zygotic mutant larvae survived within 10 days but did not survive beyond day 15, suggesting that *adnp2b* was an essential gene (Table 1, Appendix A).

The overlapping expression patterns of the two paralogues suggested that *adnp2a* and *adnp2b* may play a redundant role in brain development and function [19]. To explore this further, we went on to generate *adnp2a* and *adnp2b* double mutants. *adnp2a^−/^^−^* and *adnp2b^+/^^−^* adult zebrafish were crossed, and the resulting embryos exhibited various genotypes, reflecting the Mendel law. Notably, *adnp2a^−/^^−^*; *adnp2b^−/^^−^* embryos can be identified within 10 days but were not observed beyond day 16, consistent with the essential role of *adnp2b*. Therefore, *adnp2a^−/^^−^*; *adnp2b^−/^^−^* fish (the *adnp2* knockout line), which were larval lethal, can only be used for larval tests.

### 2.2. Histomorphological Analysis of adnpa^−/−^; adnpb^−/−^ and adnp2 Mutant Zebrafish

Compared to controls, 4-day *adnpa*^−/^^−^; *adnpb^−/^^−^* and *adnp2a*^−/^^−^; *adnp2b*^−/^^−^ larvae exhibited reduced body length and interorbital distance (Figure 1B–D). Moreover, the craniofacial deficits observed in the *adnpa*^−/^^−^; *adnpb*^−/^^−^ embryos were confirmed through alcian blue staining, a technique that binds to sulfated glycosaminoglycans present in the cartilage matrix and thereby indicates the development of cartilage in zebrafish (Figure 1E). Specifically, angle between Meckel’s cartilage and ceratohyal bone structures were increased significantly, and length of palatoquadrates were longer in *adnpa*^−/^^−^; *adnpb*^−/^^−^ embryos than in controls (Appendix A).

We asked whether microcephaly and craniofacial deficits persisted in adults. *adnpa^−/^^−^*; *adnpb^−/^^−^* adults showed incomplete operculum, and lengthened and protruded jaws (Figure 1F), as confirmed by computerized tomography (CT) scan (Figure 1G). The brain size of *adnpa^−/^^−^*; *adnpb^−/^^−^* fish remained smaller compared to controls (Figure 1H), with the cerebellum area specifically reduced to 75% of that in controls (Appendix A). The HE staining results showed that the brain structure of *adnpa^−/^^−^*; *adnpb^−/^^−^* fish had also undergone significant changes (Appendix A). In contrast, the brain size was comparable between *adnp2a^−/^^−^*; *adnp2b^+/^^−^* and controls.

### 2.3. adnpa^−/−^; adnpb^−/−^ and adnp2 Mutant Larvae Exhibit Behavioral Deficits

To investigate whether deficiency of *adnp* and *adnp2* causes the change in larval behaviors, we conducted an open field test to evaluate locomotor and thigmotactic activities of 7 dpf larvae. Both *adnpa^−/^^−^*; *adnpb^−/^^−^* and *adnp2* mutant larvae moved significantly less than controls (Figure 2A,B; Appendix A). Notably, compared to *adnp2a^−/^^−^*; *adnp2b^+/+^* and *adnp2a^−/^^−^*; *adnp2b^+/^^−^*, *adnp2a^−/^^−^*; *adnp2b^−/^^−^* larvae moved the shortest distances, suggesting a potential dose-dependent role of *adnp2b*.

Zebrafish show natural thigmotaxis to adapt to new environments. The traces of individual *adnpa^−/^^−^*; *adnpb^−/^^−^* and *adnp2* mutant larvae were obviously different from that of the controls (Figure 2C). Both *adnpa^−/^^−^*; *adnpb^−/^^−^* and *adnp2* mutant larvae spent significantly less time in the peripheral zone compared to the wild-type controls. By contrast, both *adnpa^−/^^−^*; *adnpb^−/^^−^* and *adnp2* mutant larvae spent more time in the center zone than controls (Figure 2D; Appendix A).

We also examined the response evoked by the light/dark transitions. Following a 30 min acclimation period, monitored through three 30 min light/dark cycles (Figure 2E). Light-to-dark transition elicited sudden increase in total distance traveled, and dark-to-light transition resulted in sudden decrease in total distance. *adnpa^−/^^−^*; *adnpb^−/^^−^
*larvae were less responsive to light/dark change, and moved the shortest distances, in the light condition in particular (Figure 2F). Compared to *adnp2a^−/^^−^*; *adnp2b^+/+^* and *adnp2a^−/^^−^*; *adnp2b^+/^^−^*, *adnp2a^−/^^−^*; *adnp2b^−/^^−^
*larvae moved the shortest distances in both light and dark conditions, suggesting a dose-dependent role of *adnp2b* once again.

Taken together, these findings suggest that disruption of *adnp* and *adnp2* alters the locomotor activity, thigmotaxis, and the response to light/dark shifts in zebrafish larvae.

### 2.4. adnpa^−/−^; adnpb^−/−^ and adnp2a^−/−^; adnp2b^+/−^ Adult Zebrafish Show Repetitive or Anxious Behaviors

We went on to examine the behaviors in *adnp* and *adnp2* mutant adults. First, the locomotor behaviors of adult fish were examined in an illuminated tank. The velocity and distance moved were comparable between *adnpa^−/^^−^*; *adnpb^−/^^−^* and control adults (Figure 3A,B). However, *adnp2a^−/^^−^*; *adnp2b^+/^^−^* adults displayed a reduced velocity, and moved significantly less compared to controls.

To determine whether *adnp* and *adnp2* deficiency leads to thigmotaxis change, we accessed the time spent in the center vs. in the peripheral zones. *adnpa^−/^^−^*; *adnpb^−/^^−^* and *adnp2a^−/^^−^*; *adnp2b^+/^^−^* adults showed the traces different from that of the controls (Figure 3C; Appendix A). The time spent in central zone was significantly more in *adnp2a^−/^^−^*; *adnp2b^+/^^−^* adults than in controls (Figure 3D; Appendix A). Further analysis of activity and swimming patterns in a blinded manner revealed that *adnpa^−/^^−^*; *adnpb^−/^^−^* adults displayed some stereotypical and repetitive behaviors, including the small circling, walling, cornering, and stereotypical figure “8” swimming (Figure 3E,F). By contrast, *adnp2a^−/^^−^*; *adnp2b^+/^^−^* adults showed no such stereotypical behaviors.

To evaluate the stress- and anxiety-like behaviors, the light/dark box test was performed (Figure 3G). Control zebrafish showed a preference for dark zone instead of natural light zone. In contrast, *adnpa^−/^^−^*; *adnpb^−/^^−^* zebrafish showed a significantly reduced frequency of entering the light zone and spent less time in the light zone compared to controls (Figure 3H; Appendix A). It indicated that the *adnpa^−/^^−^*; *adnpb^−/^^−^* shows altered natural responses to the light/dark environment compared to the controls, effectively assessing the anxiety behavior of the *adnpa^−/^^−^*; *adnpb^−/^^−^*. Next, the novel tank test was performed (Figure 3I). The results indicated that the time spent in bottom zone of *adnpa^−/^^−^*; *adnpb^−/^^−^
*and *adnp2a^−/^^−^*; *adnp2b^+/^^−^* were longer than that of the control group, and the time in the middle and top zones were shorter (Figure 3J; Appendix A). These findings collectively suggest that *adnpa^−/^^−^*; *adnpb^−/^^−^* and *adnp2a^−/^^−^*; *adnp2b^+/^^−^* zebrafish exhibit heightened stress- and anxiety-like behaviors relative to controls.

### 2.5. Impaired Social Preference in adnpa^−/−^; adnpb^−/−^ and adnp2a^−/−^; adnp2b^+/−^ Adult Zebrafish

Zebrafish typically swim in a schooling that reflects the social nature of the species. To investigate whether the social interaction was altered in *adnpa^−/^^−^*; *adnpb^−/^^−^* and *adnp2a^−/^^−^*; *adnp2b^+/^^−^* adults, we conducted the shoaling test (Figure 4A; Appendix A). Results showed the average inter-fish distance and contact duration were significantly different in *adnpa^−/^^−^*; *adnpb^−/^^−^
*and *adnp2a^−/^^−^*; *adnp2b^+/^^−^* fish from that of controls (Figure 4B; Appendix A). The average distance in *adnpa^−/^^−^*; *adnpb^−/^^−^
*and *adnp2a^−/^^−^*; *adnp2b^+/^^−^* fish were significantly decreased, compared to controls (Figure 4C). To further explore this, we conducted the three-tank test (Figure 4D–F; Appendix A). The three-tank test was divided into three compartments. No zebrafish were placed in the left compartments, while five conspecific zebrafish were placed in the right compartment. The central testing compartment was also divided into three zones, namely, “empty zone,” “middle zone,” and “social zone.” By comparing the time spent by different genotypes of zebrafish in each zone with the time spent interacting with conspecifics in the right compartment, we assessed changes in social behavior. The results showed that both *adnpa^−/^^−^*; *adnpb^−/^^−^* and *adnp2a^−/^^−^*; *adnp2b^+/^^−^* fish spent significantly less time in the “social zone” compared to controls and spent more time in the “middle zone” and “empty zone.” This indicated that the social behavior of both *adnpa^−/^^−^*; *adnpb^−/^^−^* and *adnp2a^−/^^−^*; *adnp2b^+/^^−^* fish were abnormal compared to the controls. To show that *adnpa^−/^^−^*; *adnpb^−/^^−^
*and *adnp2a^−/^^−^*; *adnp2b^+/^^−^* adults had abnormal aggression behaviors, a mirror test was performed. The results showed that *adnpa^−/^^−^*; *adnpb^−/^^−^
*and *adnp2a^−/^^−^*; *adnp2b^+/^^−^* adults spent less time in the mirror “contact zone” than controls, with concomitantly more time in “approach zone” (Figure 4G,H; Appendix A).

Taken together, we concluded that *adnpa^−/^^−^*; *adnpb^−/^^−^
*and *adnp2a^−/^^−^*; *adnp2b^+/^^−^* zebrafish showed impaired social interaction behaviors. *adnpa^−/^^−^*; *adnpb^−/^^−^* fish displayed some core features of ASD, including stereotypical and repetitive behaviors, impaired social interaction, anxiety-like behavior, and abnormal aggression, whereas *adnp2a^−/^^−^*; *adnp2b^+/^^−^* fish exhibited anxiety and social preference problems.

### 2.6. Transcriptome Assay for Control and Mutant Larvae

To investigate the molecular mechanism by which disruption of *adnp* and *adnp2* leads to abnormal behaviors, bulk RNA sequencing was performed for brain tissues isolated from 7 dpf larvae.

Approximately 6400 genes were found differentially expressed between control and *adnpa^−/^^−^*; *adnpb^−/^^−^* larvae (Appendix A). 4427 differentially expressed genes (DEGs) were upregulated, and 1949 were downregulated, which was in line with the fact that that Adnp functions predominantly as a transcriptional repressor [8,37]. KEGG (Kyoto Encyclopedia of Genes and Genomes) analysis of the downregulated DEGs showed enriched terms such as GABAergic synapse (including *gabrr2b*, *gabra6a*), circadian rhythm (including *cry1aa*, *per2*, *nr1d1*), apoptosis (including *atf4a*, *atf4b*), cholinergic synapse (including *gng3*, *gngt2a*), and glutamatergic synapse (including vesicular glutamate transporter (VGluT)-related genes: *slc38a2*, *slc1a8a*) (Figure 5A). GO (Gene Ontology) analysis of the downregulated DEGs showed the enriched terms such as response to stimulus, response to cadmium ion, circadian rhythm, and the estrogen biosynthetic process (Appendix A). Consistently, core circadian clock genes such as *cry1*/*2*, *per1*/*2*, and *nr1d1*; early response genes *egr2*/*4*, *fosa/b*, and *jun*; neuroendocrine genes such as *igf3*, *prl2*, *prlh*, *esr1*, *hsd11b2*, *star*, *tshb*, and *gdf3*; and neurodegenerative and neuropsychiatry genes such as *bdnf*, *chd2*, *es1*, *ngf* and *noto* were downregulated (Figure 5C). Our results were in line with previous work showing that Adnp is involved in the regulation sex-steroid biosynthesis and circadian rhythm [38].

Approximately 320 genes were found differentially expressed between control and *adnp2a^−/^^−^*; *adnp2b^−/^^−^* larvae (Appendix A). In total, 179 genes were upregulated, and 137 genes were downregulated. KEGG analysis of the downregulated DEGs showed the enriched terms such as Parkinson’s and Huntington’s diseases (Figure 5B). GO analysis of the downregulated DEGs showed the enriched terms such as axon regeneration, regulation of neuroinflammatory response, axon choice point recognition, and positive regulation of cholesterol biosynthetic process (Appendix A).

To confirm our RNA-seq results, whole-mount in situ hybridization (WISH) and qRT-PCR were performed to detect the expression of selected genes. WISH results showed that the expression of excitatory glutaminergic, inhibitory GABAergic, and dopaminergic synaptic genes were significantly decreased in *adnpa^−/^^−^*; *adnpb^−/^^−^* larval brain (Figure 5D). Specifically, genes such as *vglut1*, *vglut2a*, *vglut2b* (excitatory glutaminergic), *gad1b*, *gad2* (inhibitory GABAergic), and *th* (dopaminergic) showed reduced expression levels in *adnpa^−/^^−^*; *adnpb^−/^^−^* larvae. qRT-PCR further confirmed these findings, showing significant deregulation in the expression of key genes involved in synaptic function (*vglut1*, *vglut2a*, *vglut2b*, *gad1b*, *gad2*, and *th*), apoptosis (*bcl-2*, *caspase3*, and *caspase9*), and microtubule dynamics (*mapta*, *map6a*). Interestingly, genes related to autophagy such as *p53* and *atg5* were significantly upregulated in *adnpa^−/^^−^*; *adnpb^−/^^−^* larval brains, indicating potential compensatory mechanisms or dysregulation in cellular homeostasis (Figure 5E; Appendix A). Regardless, autophagy-related genes *atg5* and *lc3*, or apoptosis-related genes *caspase3* and *caspase9*, were increased significantly in *adnp2a^−/^^−^*; *adnp2b^−/^^−^* larval brain. Moreover, core circadian clock genes such as *cry1*/*2*, *per1*/*2*, and *nr1d1*; immediate early response genes *egr2*/*4*, *fosab*, and *jun*; steroidogenesis and neuroendocrine genes such as *prl2*, *prlh*, *hsd11b2*, and *star;* and neurodegenerative and neuropsychiatry genes such as *bdnf*, *chd2*, *es1*, and *ngf* were downregulated in *adnpa^−/^^−^*; *adnpb^−/^^−^* larval brain (Figure 5E).

The data indicated that *adnp* and *adnp2* have common and distinct roles. *adnp* is closely associated with synaptic pathway, autophagy, apoptosis, estrogen signaling, and circadian clock, while *adnp2* is more associated with neuroinflammatory response, axon regeneration, recognition, and cholesterol biosynthesis process. These distinct roles suggest specialized functions for *adnp* and *adnp2* in various aspects of neural development and function.

### 2.7. Deletion of adnp and adnp2 Results in Decreased Levels of HuC and α-Tubulin Proteins in Larval Brains

Behavior reflects the function of the nervous system. Ultimately, we employed immunofluorescence assays to examine the expression of the pan-neuronal marker HuC and the neuronal fiber network α-tubulin in the brains of 3-day-old zebrafish larvae, aiming to investigate the neurodevelopmental status at the protein level in mutant strains (Figure 6A,B). The results clearly demonstrated that both HuC and α-tubulin showed reduced red fluorescence in *adnpa^−/^^−^*; *adnpb^−/^^−^* and *adnp2a^−/^^−^*; *adnp2b^−/^^−^* compared to wild-type counterparts. Further statistical analysis confirmed that the loss of *adnp* and *adnp2* significantly decreased the fluorescence signals of HuC and α-tubulin in zebrafish head tissues (Figure 6C–F). In conclusion, our study finds that mutations in the *adnp* and *adnp2* genes impact early neurodevelopment, leading to brain functional defects in both juvenile and adult zebrafish.

### 2.8. EEDs Exposure Aggravates Behavioral Abnormalities in adnpa^−/−^; adnpb^−/−^ and adnp2a^−/−^; adnp2b^−/−^ Larvae

ADNP plays dual roles in the central nervous system, serving as both a neuroprotective and neurotrophic agent, as well as contributing to toxic protection. To investigate whether exposure to appropriate doses of EEDs can aggravate behavioral abnormalities in *adnp* or *adnp2* genetic mutant larvae, we first determined the appropriate doses of BPA and PFOS exposure. Initially, wild-type embryos were initially exposed to a wider range of PFOS (0.01, 0.1, 0.4, 1, and 5 µM) and BPA (0.5, 1, 2, 2.5, 5, and 10 mg/L). We found that >5 mg/L BPA or >1 µM PFOS exposure for 3 days caused obvious body malformations and death of the embryos (Appendix A), indicative of strong embryonic toxicity. Subsequently, when we assessed locomotor activities [34], we found that 0.01–0.1 µM PFOS and 0.1–0.5 mg/L BPA exposure led to little alteration in locomotor activity, whereas slightly higher doses of PFOS (0.4 and 1 µM) and BPA (1 and 2 mg/L) induced noticeable changes (Appendix A). Finally, we utilized a *gad1*-RFP reporter line, which labels GABAergic neurons in regions such as subpallium, thalamus, ventral hypothalamic zone, tectumopticum, mesencephalon, and rhombencephalon with RFP [39]. Examination of RFP signals showed that prolonged lower doses of EEDs exposure did not significantly alter in neurogenesis in brain (Appendix A). Based on the above results, we selected 0.1 µM PFOS and 0.5 mg/L BPA for subsequent experiments.

Next, wild-type and mutant embryos were exposed to low doses of EEDs, and the resulting larvae at 7 dpf were subjected to behavioral assays to detect locomotor activity and the response to light/dark changes. Under continuous illumination conditions, 0.1 µM PFOS exposure had no detectable effect on traveled distances of wild-type and *adnpa^−/^^−^*; *adnpb^−/^^−^* larvae (Figure 7A,B); however, it caused an increase in the distance travelled by *adnp2a^−/^^−^*; *adnp2b^−/^^−^* larvae. Under light/dark change conditions, 0.1 µM PFOS exposure led to increased distances travelled by both *adnpa^−/^^−^*; *adnpb^−/^^−^* and *adnp2a^−/^^−^*; *adnp2b^−/^^−^* larvae but had no effects on wild-type controls (Figure 7C,D).

Under continuous illumination conditions, 0.5 mg/L BPA exposure had little effect on travelled distance of wild-type larvae; however, it led to a decrease in distances travelled by *adnpa^−/^^−^*; *adnpb^−/^^−^* larvae (Figure 7E,F). Under light/dark change conditions, BPA exposure had little effects on wild-type and *adnp2a^−/^^−^*; *adnp2b^−/^^−^* larvae (Figure 7G,H). By contrast, BPA exposure led to decreased distances travelled by *adnpa^−/−^*; *adnpb^−/^^−^* larvae in the first light/dark cycle, but no change in the remaining two cycles.

The above data suggest that low-dose EED exposure in general can aggravate behavioral abnormalities in both *adnpa^−/^^−^*; *adnpb^−/^^−^* and *adnp2a^−/^^−^*; *adnp2b^−/^^−^* larvae, in locomotor activity and response to light/dark tests.

## 3. Discussion

### 3.1. adnp and adnp2 Mutants Show Core Phenotypes of NDDs

*ADNP* family members are risk factors for NDDs. *ADNP* is a high-confidence ASD gene, whereas the *ADNP2* gene is poorly studied but closely linked with SZ/PTSD. Intensive molecular and functional assays for ADNP have been reported, which have greatly facilitated the understanding of the etiology of the diseases. In mice, *Adnp* heterozygous mutants have been generated, and they exhibit cognitive and social impairments, developmental delays, and abnormal synapses [17]. Despite these advances, the precise molecular mechanisms underlying these phenotypes remain incompletely understood. Zebrafish have been widely used in modeling human diseases and in dissecting the underlying mechanisms [22]. In this work, we have two main goals: Firstly, we would like to investigate whether the *adnp* and *adnp2* zebrafish mutants display neural behavioral phenotypes that mimic human patients, therefore establishing zebrafish disease models. Secondly, we seek to explore the causal relationship between *adnp* family gene deficiency and NDDs through a comprehensive investigation spanning morphological, molecular, and behavioral analyses.

*adnpa^−/^^−^*; *adnpb^−/^^−^* larvae showed developmental delay, craniofacial defects, and abnormal brain development. Moreover, neuronal gene expression was decreased in *adnp* mutant brain. Importantly, these larvae moved significantly less and showed abnormal thigmotaxis, and *adnpa^−/^^−^*; *adnpb^−/^^−^* adults showed impaired social preference and stereotypical and repetitive swimming behaviors, the hallmark of ASD. Thus, these findings indicate that *adnpa^−/^^−^*; *adnpb^−/^^−^* display molecular and morphological phenotypes that very well mimic those of human patients. We concluded that the zebrafish *adnp* mutant ASD model was successfully established.

We showed for the first time that *adnp2* is an essential gene in zebrafish, and its deficiency can lead to an abnormal nervous system, developmental delay, and abnormal brain development. Consistently, *adnp2* mutant fish displayed anxiety, stress, and social preference problems, similar to the *adnp* mutant. However, the *adnp2* mutant showed no stereotypical and repetitive swimming behaviors, indicating that *adnp2* is not likely a strict ASD gene. In fact, genome-wide sequencing and human genetic assay have identified *ADNP2* as a candidate SZ/PTSD gene. In the future, it will be important to investigate whether the *adnp2* mutant zebrafish show phenotypes that mimic SZ/PTSD-like behaviors.

### 3.2. Mechanistic Explanation for adnp/adnp2-Related Phenotypes

The availability of *adnp* and *adnp2* mutant zebrafish allowed for us to analyze the relationship between transcriptome change in brain and neural behavioral alterations.

Dendritic spine is the main site of excitatory synaptic input, and its structure and function are abnormal in *Adnp^+/^^−^* mice [21,40]. Our qRT-PCR and WISH results showed that excitatory, inhibitory, and dopaminergic synaptic genes were significantly downregulated in *adnpa^−/^^−^*; *adnpb^−/^^−^* brain. ADNP/NAP are known to regulate tau or other microtubule end-binding proteins (Maps) by binding to microtubule end-binding protein 3 (EB3) to jointly ensure neuronal survival and function [18,41,42,43,44]. *mapta* and *map6a* were significantly downregulated in *adnpa^−/^^−^*; *adnpb^−/^^−^
*and *adnp2a^−/^^−^*; *adnp2b^−/^^−^*. In *adnpa^−/^^−^*; *adnpb^−/^^−^* larvae, the expression of apoptosis genes, including *bcl-2*, *caspase3*, and *caspase9,* was decreased, while *p53* expression was increased. In *adnp2a^−/^^−^*; *adnp2b^−/^^−^
*larvae, the expression of autophagy and apoptosis genes, including *atg5, lc3*, *caspase3*, and *caspase9,* was significantly increased.

A set of steroidogenic and neuroendocrine genes, including the growth and development-regulating gene *fosab*, the estrogen synthesis gene *cyp191a1*, the neuropeptide gene *galn*, the corticosteroid gene *hsd11b2*, the steroidogenesis gene *star*, the insulin-like three-peptide gene *insl3*, and the progesterone receptor gene *pgr*, were downregulated in *adnpa^−/^^−^*; *adnpb^−/^^−^* larvae. NDDs such as ASD are known to have a sex-biased prevalence rate, and the relationship between fetal estrogens and ASDs has been observed [36,45]. ASD patients number more in males than in females, and it is suggested that female factors can reduce the likelihood of autism. Brain tissues, including hypothalamus and cortical neurons, where it has important regulatory functions in different processes such as cognition and anxiety, express estrogen and androgen receptors, and a role for sex steroids (e.g., estrogen, testosterone) in regulating neurogenesis and emerging behaviors [38,46]. Fetal estrogens play a key role in synaptogenesis and corticogenesis during early embryogenesis. In fact, *ADNP* has been shown to be sexually regulated, expressed in the hypothalamus, and may have a sex-specific role [5,47,48]. Our results suggest that Adnp is important for proper steroid levels by regulating steroidogenesis and neuroendocrine genes in brain tissues.

In *adnpa^−/^^−^*; *adnpb^−/^^−^* larvae, circadian clock genes such as *per1*/*2*, *nr1d1*/*2*, and *cry1*/*2* were downregulated, suggesting that the circadian clock might be affected. In fact, ASD children show circadian rhythm problems and sleep disturbances. Circadian rhythm components can modulate aggressive behavior, and altered clock gene expressions have been associated with NDDs [49]. Our data thus suggest that disruption of the circadian clock by ADNP dysfunction may contribute to ASD [50]. We also found that early response genes, such as *egr2*/*4*, *fosab*, and *jun*, which are rapidly and transiently expressed in response to cellular stimuli, play crucial roles in neurodevelopment, memory formation, and stress responses [51,52]. These genes are downregulated in *adnp* mutant larvae. Similarly, genes associated with neurodegenerative and neuropsychiatric disorders, such as *bdnf*, *chd2*, *es1*, *ngf*, and *noto*, are also downregulated in *adnp* mutants. For instance, *bdnf* (brain-derived neurotrophic factor) is involved in the growth and development of glutamatergic and GABAergic synapses and regulates dopaminergic neurotransmission. Abnormal expression of *bdnf* is linked to major diseases such as Huntington’s disease, Alzheimer’s disease, schizophrenia, and anxiety disorders. *noto* plays a crucial role in early embryonic development, affecting the formation and function of the nervous system [53,54].

Overall, the downregulated KEGG and GO results from RNA-seq indicate that zebrafish lacking *adnp* exhibit a reduction in several critical pathways involved in cellular metabolism, protein synthesis, neural development, apoptosis, and immune response [10,40,55,56]. The observed decrease in cerebellar size and behavioral abnormalities in these mutant zebrafish can be attributed to disruptions in these pathways, which impact neuron proliferation, differentiation, and function, leading to cerebellar underdevelopment or atrophy. Specifically, the affected pathways include those related to cell proliferation (e.g., MAPK and ribosome pathways), neural development (e.g., GABAergic and glutamatergic synapse pathways), metabolic regulation (e.g., retinol metabolism and oxidative phosphorylation pathways), and immune and inflammatory responses (e.g., antigen processing and IL-17 signaling pathways). Conversely, zebrafish lacking *adnp2* primarily influence development and differentiation pathways, which directly impact cerebellar development and cell differentiation processes (e.g., estrogen signaling pathway, ECM–receptor interaction, steroid biosynthesis, focal adhesion, and PI3K-Akt signaling pathway) [57], as well as metabolic regulation pathways (e.g., protein digestion and absorption; biotin metabolism; glycine, serine, and threonine metabolism; and thermogenesis). Additionally, pathways related to cellular function and health are also affected (including oxidative phosphorylation, apoptosis, Parkinson’s disease, Huntington’s disease, and non-alcoholic fatty liver disease). These disruptions impact cellular function, health, and stress responses, further influencing cerebellar development. In summary, transcriptomic data suggest that the absence of *adnp* and *adnp2* in zebrafish affects neuron proliferation, differentiation, and function, which may be the primary mechanisms leading to cerebellar underdevelopment, atrophy, and behavioral changes.

### 3.3. EEDs Aggravate Neural Behavioral Phenotypes

It is generally believed that gene–environment interactions contribute to a portion of the phenotypic variation in NDDs [58]. In this work, we showed that prolonged low-dose EED exposure can aggravate neural behavioral phenotypes in NDD-risk genetic mutant zebrafish. However, different EEDs displayed different effects depending on the mutant background. BPA appeared to have stronger effects on *adnp* mutant, while PFOS seemed to have stronger effects on *adnp2* mutant. These observations may interpret the heterogeneity and complexity of NDDs, provided that the gene–environmental interaction mechanism is truly at work.

The molecular mechanisms by which EEDs synergize with mutations of NDD-risk genes for neural and behavioral outcomes were not investigated in this work. Human neurodevelopmental diseases are well linked to epigenetic disruptions. NDD-related factors such as ADNP, ADNP2, and POGZ are known chromatin regulators that can modulate histone modification or DNA methylation at genome-wide [6,7,8], and environmental factors or pollutants can induce epigenomic change [34,59]. We speculate that EEDs and genetic factors may synergize to affect the expression of genes that are involved in brain function by altering the epigenome. With genetic mutant mouse embryonic stem cells and zebrafish in hand, our lab is currently investigating this.

## 4. Methods

### 4.1. Zebrafish Maintenance

The AB strain of zebrafish was used in this work. Both the wild type and the mutants were raised in the same conditions, with the temperature at 28.5 °C and the PH at about 7. The light was turned on at 9:00 a.m. and turned off at 11:00 p.m., therefore with 14 h light/10 h dark.

All zebrafish experiments followed the principle of the Institutional Animal Care and Use Committee of the Institute of Hydrobiology, Chinese Academy of Sciences, under the number IHB2014-006.

### 4.2. Alcian Blue Staining

Alcian blue staining protocol has been described previously [8]. In brief, the embryos were fixed at 4 °C overnight with 4% paraformaldehyde. Next day, embryos were dehydrated with 50% ethanol for 10 min, and then stained with 0.2 mg/mL alcian blue 8 GX (33864-99-2, AMRESCO, Solon, OH, USA) in 70% ethanol/80 mM MgCl_2_ at room temperature.

### 4.3. Generation of adnp2 Mutants by CRISPR/Cas9

Zebrafish *adnp2* mutants were generated using CRISPR/Cas9 technology. We designed gRNAs against *adnp2a* and *adnp2b*, targeting exon 2 of the *adnp2a* gene, and exon 4 of the *adnp2b* gene. The gRNA targeting sequences for *adnp2a* and *adnp2b* were 5′-GGACTCAGACGACCGAGGAAAC-3′ and 5′-GGGGTGGGCTATAAACGGGC-3′, respectively. gRNAs were generated using the MEGA shortscript T7 kit (Thermo Fisher Scientific, Waltham, MA, USA). The Cas9 mRNA was synthesized using the mMESSAGE mMACHINE SP6 Kit (AM1340, Thermo Fisher Scientific, Waltham, MA, USA). The mixture containing 200 ng/µL Cas9 mRNA and 80 ng/µL gRNA was co-injected into 1-cell stage zebrafish embryos. The genomic DNA of 20 injected embryos at 24 hpf was extracted and subjected to PCR amplification. The DNA fragments containing the gRNA targeting sequences were amplified by PCR using primers flanking the targeting sites of the *adnp2a* and *adnp2b* genes. The primers for *adnp2a* are 5′-GACGCGCGCAGACATTTATC-3′ (forward) and 5′-GCTGGAGGGGCTGATTTGTAA-3′ (reverse), and for *adnp2b* are 5′-AGTGGGAATATCGGACACAAGG-3′ (forward) and 5′-GGTACAGCAAGTGTTCGGATG-3′ (reverse). The genotype was determined by DNA sequencing. Adults bred from the injected embryos were screened for mosaic founders by the amplicon sequencing. The mosaic founders were outcrossed to wild type to obtain the F1 offspring with stable germline transmission. The F1 heterozygotes were outcrossed to wild type to generate F2 heterozygotes. The F2 *adnp2a* heterozygotes were inter-crossed, resulting in homozygous mutants. No *adnp2b* homozygous embryos were found viable at and after 15 dpf, suggesting that *adnp2b* is an essential gene. The *adnp2a^−/^^−^* and *adnp2b^+/^^−^* adults were outcrossed to generate the double heterozygotes. When double heterozygous adults were crossed, *adnp2a^−/^^−^*; *adnp2b^+/^^−^* embryos (no *adnp2a^−/^^−^*; *adnp2b^−/^^−^
*adults were viable), which had the least gene dosage of *adnp2* genes, were kept and raised up to adulthood for behavioral analysis. The genotypes were determined by PCR and electrophoresis analyses.

### 4.4. Locomotor Activity and Thigmotaxis Tests for Larvae

All behavioral tests were conducted between 10:00 a.m. and 4:00 p.m. The tests were conducted and analyzed by the EthoVision XT 15 software (Noldus, Wageningen, The Netherlands), with the camera resolution at 1280 × 960 pixels, the frame rate at 25/s, and the tracking feature as center point detection. Experiments were performed in a 24-well plate, and each well was regarded as an observation area. For thigmotaxis test, the inner concentric circle was set as “center zone” and the outer one was “periphery zone”.

The day before the test, individual zebrafish larvae were placed in each well of the 24-well plate. The next day, after 40 min of acclimation, fish were recorded for 20 min in continuous illumination, and then 30 min for 3 light/dark cycles (5 min dark, 5 min light; 5 min dark, 5 min light; 5 min dark, 5 min light) [24].

### 4.5. Open-Field and Thigmotaxis Tests for Adults

All the adults used for behavioral tests were males of 3.5 to 4 months old. Behaviors were documented using Zebrafish behavior analysis system (Zeb-View, Almere, NL, The Netherlands), and analyzed by the EthoVision XT 15 software (Noldus, The Netherlands). The tank size for the open-field and thigmotaxis tests was 20 cm × 20 cm × 10 cm. Before the tests, the zebrafish were acclimated for 30 min, then recorded for 30 min [60]. The stereotyped or repetitive behaviors were based on recordings made every 15 s. For the thigmotaxis test, the tank was divided into two zones: a peripheral and a central zone. The time ratio was the time spent in the peripheral zone divided by the total time spent in the tank.

### 4.6. Shoaling Test

The tank used for shoaling test was the same as one used in the open-field test. Five adult male fish were placed in the center of the tank and used for experiments. After 5 min of adaptation in a new tank, fish were recorded for a total of 15 min by a camera from above. Shoaling test analysis was based on recordings made every 30 s, and the inter-fish distance was measured.

### 4.7. Social Preference Test (Three-Tank Test)

The fish were acclimated for 2 min before the test, and a total of 10 min were recorded for the social preference test. The size of the tank was 40 cm × 15 cm × 15 cm. The tank was partitioned into three compartments: a 20 cm × 15 cm × 15 cm middle compartment, and two 10 cm × 15 cm × 15 cm compartments on the left and right sides. Five conspecifics were placed in the right compartment, and zero zebrafish in the left compartment served as control. The middle compartment was divided into 3 sectors, from left to right as “empty zone”, “middle zone”, and “social zone”. The behaviors were quantified as a distance distribution in a zone adjust to the conspecifics. The time ration was the time spent in the conspecific sector divided by the total time.

### 4.8. Novel Tank Test

Before the experiment, the fish were acclimated for 5 min and then recorded for 15 min by a camera from above. The size of the tank used was 28 cm × 20 cm × 5 cm, and was divided into three equal compartments, from top to bottom as “top zone”, “middle zone”, and “bottom zone”.

### 4.9. Light/Dark Test

The size of the fish tank was 40 cm × 10 cm × 10 cm. The tank was divided into two: light area and dark area. After 2 min acclimation, the fish were recorded for 10 min by a camera from the top. The data in the light area were used for the analysis.

### 4.10. Mirror Test

The mirror and the fish tank were placed with an angle of 22.5°. The size of the tank was 30 cm × 10 cm × 15 cm and was divided into 3 areas: “mirror contact zone”, “approach zone”, and “far zone”. Individual fish were acclimated for 2 min and recorded for 12 min by a camera from the top.

### 4.11. RNA-Seq Analysis

The brain tissues from 50 larvae at 7 dpf were collected for total RNA extraction and used for RNA sequencing. Total RNA was isolated using the Trizol reagent (Thermo Fisher Scientific, Waltham, MA, USA). RNA sample quality was checked by the OD 260/280 ratio using the Nanodrop 2000 instrument. The RNA samples were sent to BGI company, China, where the libraries were constructed, and sequencing by a BGI-500 system. RNA-seq experiments were performed in at least two replicates.

The RNA-seq data were aligned to the zebrafish reference genome using HISTA2. Then, raw counts of all protein coding genes were generated by FeatureCounts. Raw counts were normalized to TPM (transcript per million). DESeq2 was performed to calculate differentially expressed genes with abs|log2(fold change) > 1 and *p*-adj < 0.05. ggplot2 was used to make the scatter plot. For heat map analysis, TPM was used, and the plot was made by heatmap. Differentially expressed genes (DEGs) were defined by FDR < 0.05 and a Log2 fold change > 1. Gene Ontology (GO) analysis for differentially expressed genes (DEGs) was performed at https://geneontology.org (accessed on 23 November 2023).

### 4.12. In Situ Hybridization

Briefly, 1-Phenyl-2-thiourea (PTU, 0.0045%, Sigma, P7629, St. Louis, MO, USA) was added to inhibit pigment, and 4-day-old larvae were fixed in 4% paraformaldehyde at 4 °C overnight. Gradient dehydration to 100% methanol was performed and stored at −20 °C overnight. After rehydration and digestion by protease K, embryos were fixed in 4% paraformaldehyde for 20 min. After pre-hybridization in a 70 °C water bath, the anti-sense DIG-labeled RNA probes were added, and incubated overnight. After washing and blocking, anti-DIG secondary antibody (Roche, 11093274910, Basel, Switzerland) was added overnight at 4 °C.

### 4.13. Immunofluorescence (IF) Experiments

Anesthetic zebrafish were fixed with 4% PFA and blocked using 5% normal bovine serum (A2153m Sigma, St. Louis, MO, USA) in PBS + 0.1% Triton-X100 (V900502, VETEC, London, UK) for 2 h. After wash, primary antibodies anti-tubulin (1:200, T7415, Sigma) and HuC (1:200, A21271, Invitrogen, Carlsbad, CA, USA) were added. Embryos were washed 3 times with PBS for 5 min; Alexa 488 or Alexa 555 (A11008, A21428, Invitrogen) secondary antibodies (1:500) were added for 2 h. The samples were counter-stained with DAPI (D9542, Sigma, St. Louis, MO, USA) in 1×PBS at room temperature for 1 h. After 3 times washing with PBS, samples were mounted using anti-fade mounting medium (SI103-02, SEVEN, Suzhou, Jiangsu, China), and imaged with a Leica Confocal Microscope (TCS SP8 STED, Mannheim, Germany).

### 4.14. BPA and PFOS Treatment

A total of 150 embryos after shield stage were collected and transferred into tanks with 500 mL egg water with appropriate concentration of BPA or PFOS. The embryos were exposed to EEDs for a total of 7 days. Every day, half the volume of an egg of water was changed, and the amount of BPA or PFOS was added to ensure the constant concentration of EEDs. DMSO was added and served as controls. The experiments were repeated at three times.

To determine the doses of BPA and PFOS, we initially tried a wider range of PFOS (0.01, 0.1, 0.4, 1, and 5 μM) [61] and BPA (0.1, 0.5, 1, 2.5, 5, 10 mg/L) [62]. By examining the embryos at 72 h, we excluded the doses that caused obvious morphological changes. Then, we utilized a locomotor behavioral test to further narrow down the doses of EEDs. A lower-dose range of EEDs, which had no effects on behaviors in wild type, was used for further experiments.

### 4.15. Statistical Analysis

All experiments were performed with three or more replicates. One-way analysis of variance (ANOVA) was used for all experiments except for line graphs and corresponding bar graphs, which utilized two-way ANOVA. Data analysis was conducted using Prism 8.0 software (GraphPad Software Inc., San Diego, CA, USA). A significance level of *p* < 0.05 was considered statistically significant. Values are presented as mean ± SEM.

## 5. Future Directions and Limitations

This study aims to investigate the molecular mechanisms of neurodevelopmental disorders associated with the loss of *adnp* family genes in zebrafish, focusing primarily on their functions in brain neurons. However, the absence of a head-specific conditional knockout may introduce additional effects from the loss of *adnp* family genes in other body regions. Additionally, our study did not delve into the single-cell level effects of *adnp* and *adnp2* loss on brain neural system function, limiting the depth of our findings.

We developed a model of *adnp* family neurodevelopmental disorders in zebrafish and examined the molecular changes following the loss of *adnp* family genes. We hope that this model will be useful in future treatments for *adnp* family deficiencies or ASD and will provide a platform for large-scale drug screening for neurodevelopmental diseases caused by *adnp* family gene loss.

## 6. Conclusions

*adnp*/*adnp2* mutant zebrafish mimic symptoms of *ADNP*/*ADNP2* patients. The neuronal markers involved include *vglut1*, *vglut2a*, *vglut2b*, *gad1b*, *gad2*, *th*, *bdnf*, and *noto*; autophagy-related genes include *p53*, *atg5*, and *atg3*; apoptosis-related genes include *bcl-2* and *caspase3*; circadian clock genes include *cry1*, *cry2*, *per1*, *per2*, and *nr1d1*; immediate early response genes include *egr2*, *egr4*, and *jun*; and estrogen signaling pathways include *igf3*, *prl2*, *galn*, *cyp19a1a*, *hsd11b2*, *star*, *insl3*, and *pgr*. These factors may mediate the development of neural disorders in *adnp*/*adnp2*. By administration of PFOS and BPA, we show that prolonged low dose of EED exposure aggravates abnormal behaviors in NDD-risk gene mutants, which provide new insights into how gene–environmental interaction may play roles in neurodevelopmental diseases.

## Figures and Tables

**Figure 1 ijms-25-09469-f001:**
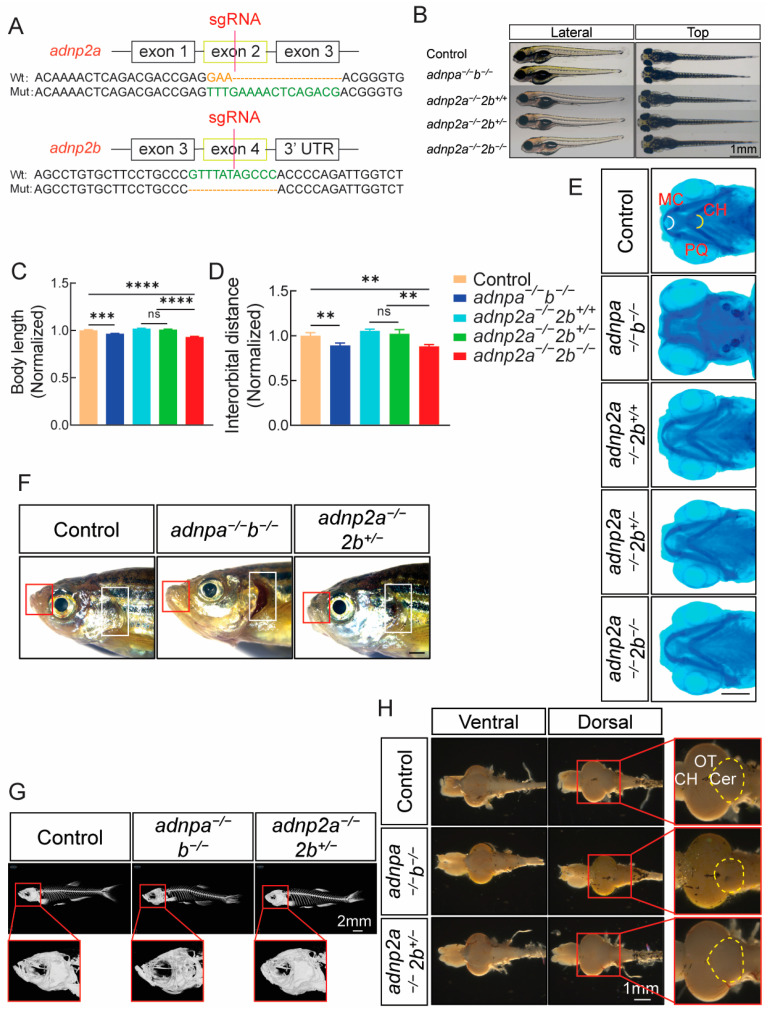
Generation and characterization of *adnp2* mutants. (**A**) The design of CRISPR/Cas9 gRNA targeting and the genotyping results. Upper: the *adnp2a* gene; lower: the *adnp2b* gene. (**B**) Morphology of the larvae of the indicated group at 4 dpf. Lateral: lateral view; top: dorsal view. Bar = 1 mm. (**C**,**D**) Relative body length and interorbital distance based on the panel B (n = 20 per group). Data shown were normalized, and the control was set as 1. (**E**) Alcian blue staining images of the larvae of the indicated groups at 4 dpf, showing the angle of Meckel’s cartilage (n = 20 per group). MC: Meckel’s cartilage; PQ: palatoquadrate; CH: ceratohyal cartilage. Bar = 0.2 mm. (**F**) Morphology of control, *adnpa^−/^^−^*; *adnpb^−/^^−^* and *adnp2a^−/^^−^*; *adnp2b^+/^^−^* adult zebrafish (at 4 mpf, all male). The red box showed the protruded lips, and the white box showed the deficits of the gills (n = 27 per group). Bar = 1 mm. (**G**) CT images (n = 6 per group) showed the head of control, *adnpa^−/^^−^*; *adnpb^−/^^−^*, and *adnp2a^−/^^−^*; *adnp2b^+/^^−^* adults. Bar = 2 mm. (**H**) Anatomical brain images of control, *adnpa^−/^^−^*; *adnpb^−/^^−^*, and *adnp2a^−/^^−^*; *adnp2b^+/^^−^* adults. Ventral: ventral view; dorsal: dorsal view. The dotted yellow lines indicated the area of the cerebellum. CH: cerebral hemisphere; OT: optic tectum; Cer: cerebellum. Bar = 1 mm. Data are presented as mean ± SEM; ** *p* < 0.01, *** *p* < 0.001, **** *p* < 0.0001. ns: no significance.

**Figure 2 ijms-25-09469-f002:**
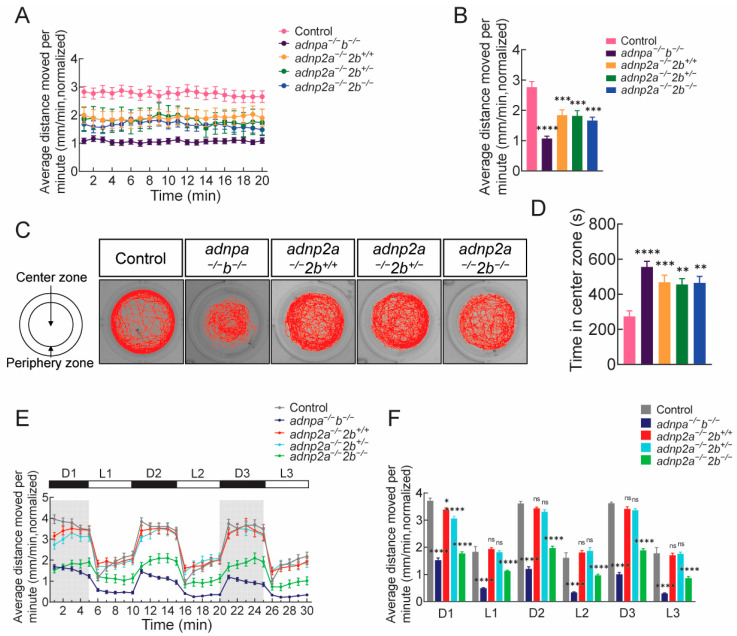
Impaired locomotor activity of mutant larvae in the open field test. (**A**) Locomotor activity of indicated larvae at 7 dpf (n = 68 per group). The average distance moved within each 1 min bin under constant illumination condition was plotted. The Y axis shows the normalized distance (millimeters) travelled by larvae in each 1 min bin. (**B**) Quantification of panel A. (**C**) Representative swimming trajectories of individual larvae of the indicated groups in the thigmotaxis test. The left side is a diagram of the observation area, the inner concentric circles are the “center zone”, and the outer concentric circles are the “periphery zone”. (**D**) Graph showing the time (second) in center zone spent by the indicated larvae types. (**E**) Light/dark test of the indicated groups at 7 dpf (n = 48 per group). The activity was recorded for 20 min, after 40 min of acclimation. Shown were based on a 30 min light/dark test, with three-min light/dark cycles (D1/L1, D2/L2, and D3/L3). D1: dark interval 1; L1: light interval 1. The average distance moved within each 1 min bin under light or dark conditions was plotted. Grey color was used to highlight the dark intervals. (**F**) Quantification of panel E. The Y axis shows the normalized distance (millimeters) travelled by larvae in each 1 min bin. Data are presented as mean ± SEM; * *p* < 0.05, ** *p* < 0.01, *** *p* < 0.001, **** *p* < 0.0001. ns: no significance.

**Figure 3 ijms-25-09469-f003:**
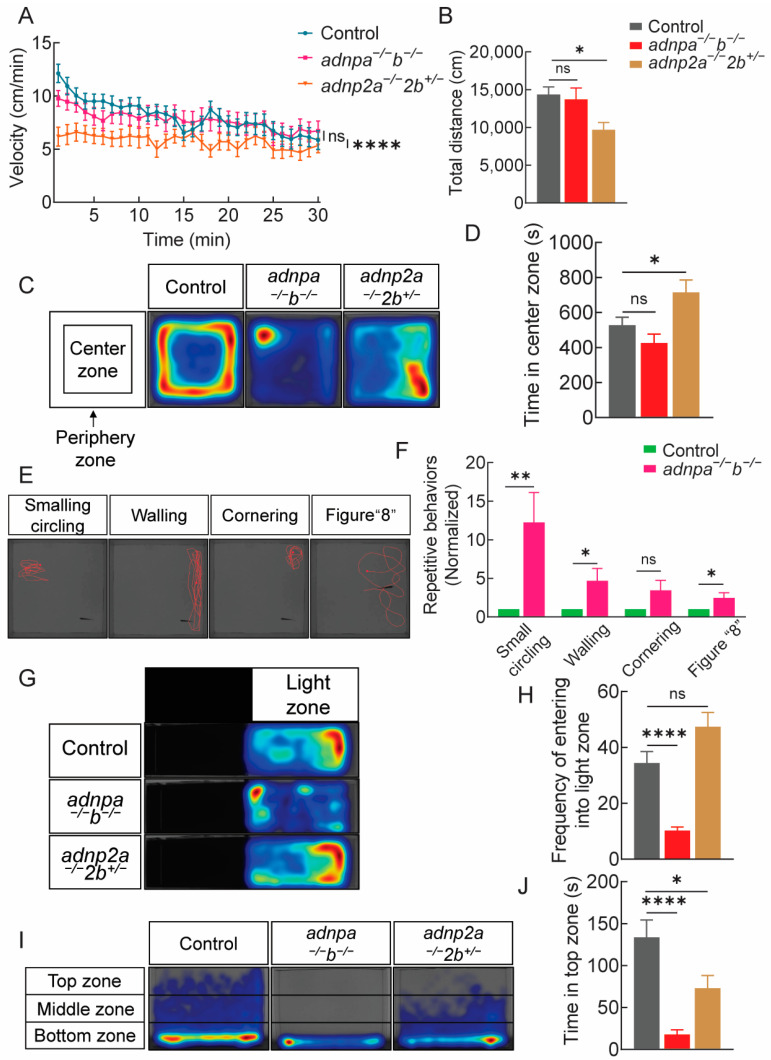
Abnormal locomotor and thigmotaxis activities of adult mutants. (**A**) Graph showing the average velocity of individual control, *adnpa^−/^^−^*; *adnpb^−/^^−^*, and *adnp2a^−/^^−^*; *adnp2b^+/^^−^* adults (n = 28 per group) at 4 mpf, in a total 30 min period. (**B**) Graph showed the average total distance travelled by control, *adnpa^−/^^−^*; *adnpb^−/^^−^*, and *adnp2a^−/^^−^*; *adnp2b^+/^^−^* adults. (**C**) The representative traces of movements of control, *adnpa^−/^^−^*; *adnpb^−/^^−^*, and *adnp2a^−/^^−^*; *adnp2b^+/^^−^* adults. (**D**) Graph showing the time (second) spent in center zone of the indicated groups. (**E**) Representative swimming trajectories showing the stereotypical and repetitive behaviors of *adnpa^−/^^−^*; *adnpb^−/^^−^
*adults. (**F**) quantification of panel (**E**). (**G**) Heat map visualization of zebrafish trajectories in the light/dark box test, in a total of 10 min period (n = 26 per group). (**H**) Graph showing the frequency of entering into light zone. (**I**) Heat map visualization of zebrafish trajectories in the novel tank test, in 10 min period (n = 30 per group). (**J**) Graph showing time spent in top zone of the indicated groups. Data are presented as mean ± SEM; * *p* < 0.05, ** *p* < 0.01, **** *p* < 0.0001. ns: no significance.

**Figure 4 ijms-25-09469-f004:**
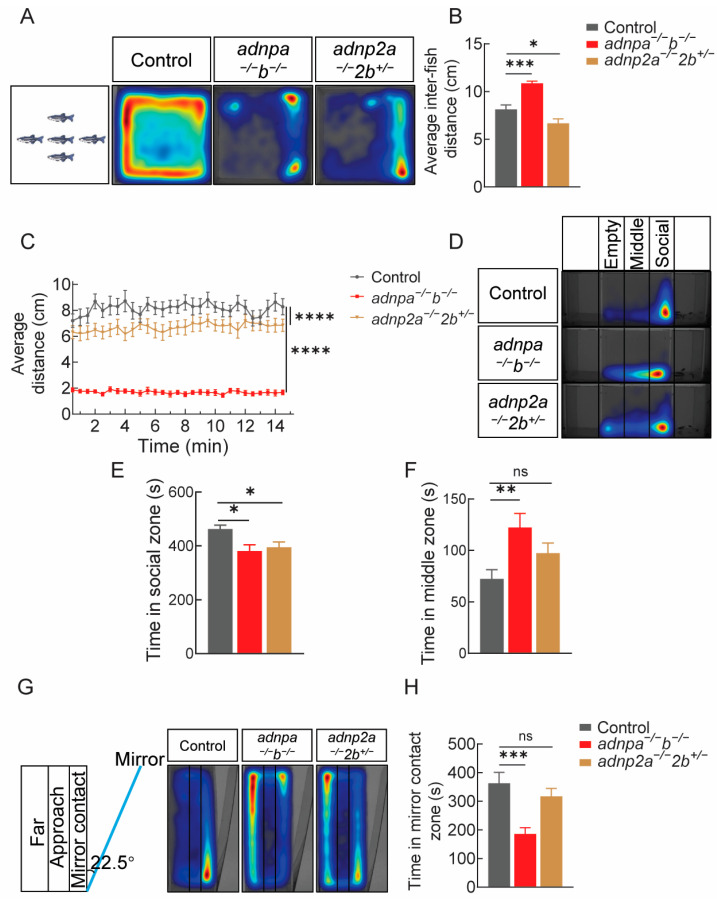
Impaired social preference behaviors in adult mutants. (**A**) Heat map visualization of zebrafish trajectories in the shoaling test of 4 mpf adults (12 groups, 5 zebrafish per group), based on 15 min. (**B**) Graph showing the average inter-fish distance. (**C**) Diagram showing the average distance moved by each group. (**D**) Heat map visualization of zebrafish trajectories in the three-tank test, in a total 10 min period (n = 26 per group). (**E**,**F**) Graph showing the time (second) spent in social and middle zones. (**G**) Heat map visualization of zebrafish trajectories in the mirror test (n = 25 per group). (**H**) Graph showing the time (second) spent in mirror contact zone. Data are presented as mean ± SEM; * *p* < 0.05, ** *p* < 0.01, *** *p* < 0.001, **** *p* < 0.0001. ns: no significance.

**Figure 5 ijms-25-09469-f005:**
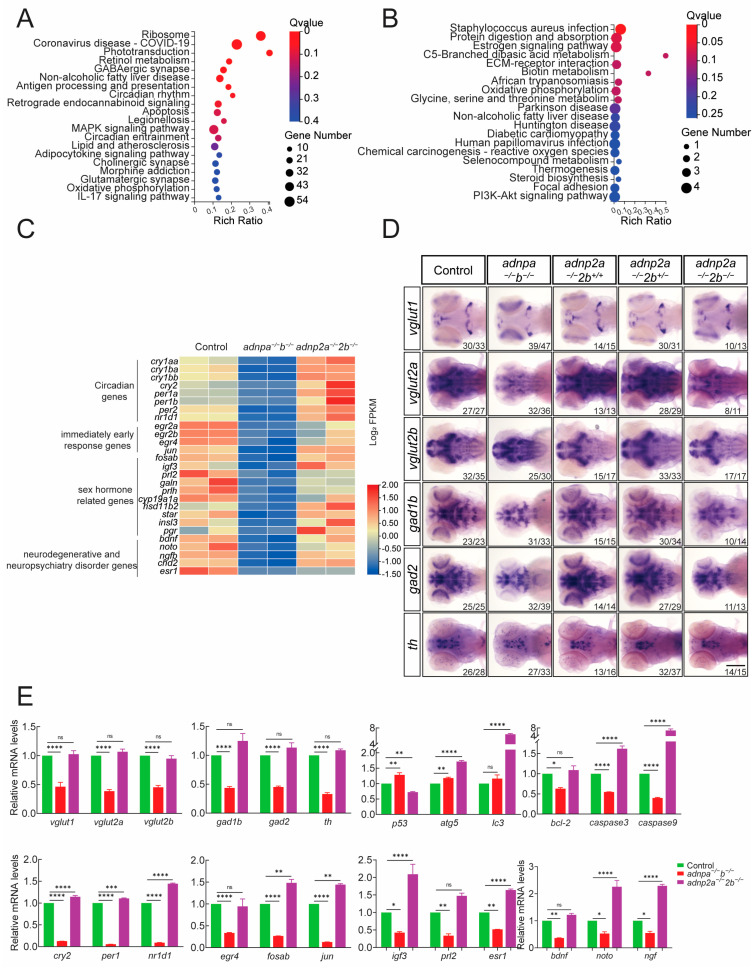
Differentially expressed genes identified by RNA-seq analysis. (**A**) KEGG terms of the downregulated DEGs of control and *adnpa^−^^/^^−^*; *adnpb^−^^/^^−^* larval brain at 7 dpf. (**B**) KEGG terms of the downregulated DEGs of control and *adnp2a^−^^/^^−^*; *adnp2b^−^^/^^−^* larval brain at 7 dpf. (**C**) Cluster heat map of circadian rhythm, early response, sex hormones, and neural regeneration pathways. (**D**) Results of in situ hybridization for synaptic-associated genes. (**E**) Corresponding to the qRT-PCR results in Figure (**C**,**D**). Data are presented as mean ± SEM; * *p* < 0.05, ** *p* < 0.01, *** *p* < 0.001, **** *p* < 0.0001, ns: no significance.

**Figure 6 ijms-25-09469-f006:**
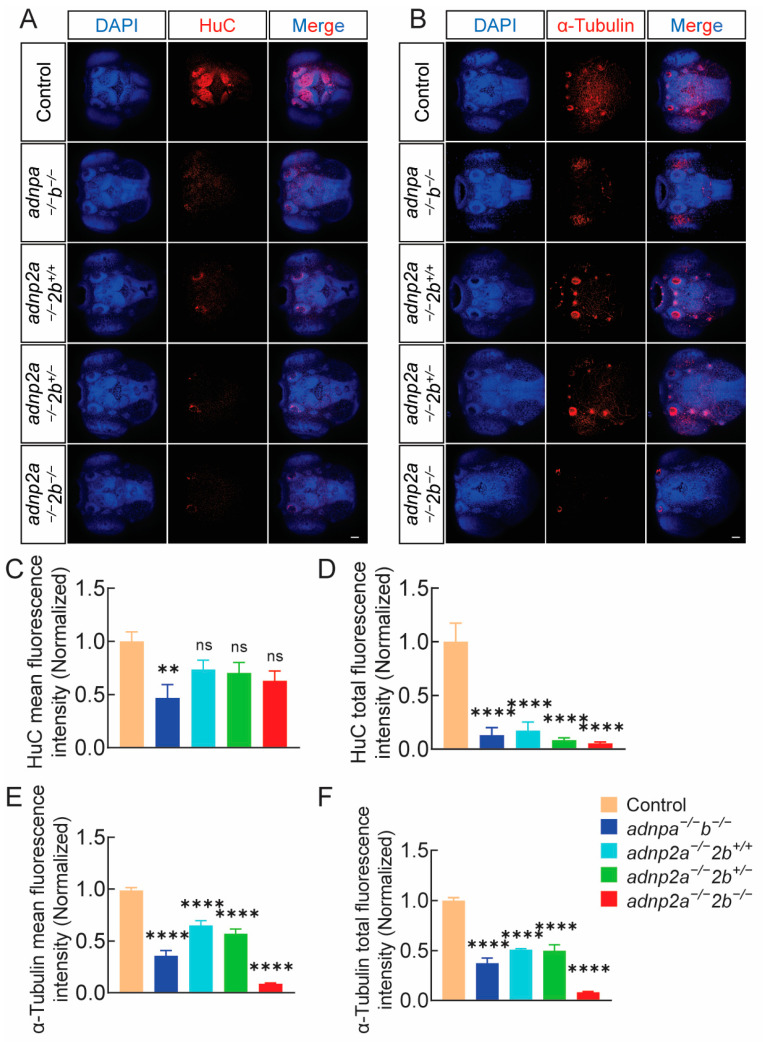
The levels of HuC and α-tubulin proteins in *adnp* and *adnp2* mutant larvae were significantly decreased. (**A**) HuC IF results in larval brains. (n = 11 per group). (**B**) α-tubulin IF results in larval brains (n = 9 for each genotype); 3 days; bar = 200 μm. (**C**,**D**) Mean fluorescence intensity and total fluorescence intensity of HuC red fluorescence signal. (**E,F**) Mean fluorescence intensity and total fluorescence intensity of α-tubulin red fluorescence signal. Data are presented as mean ± SEM; ** *p* < 0.01, **** *p* < 0.0001, ns indicates no significance.

**Figure 7 ijms-25-09469-f007:**
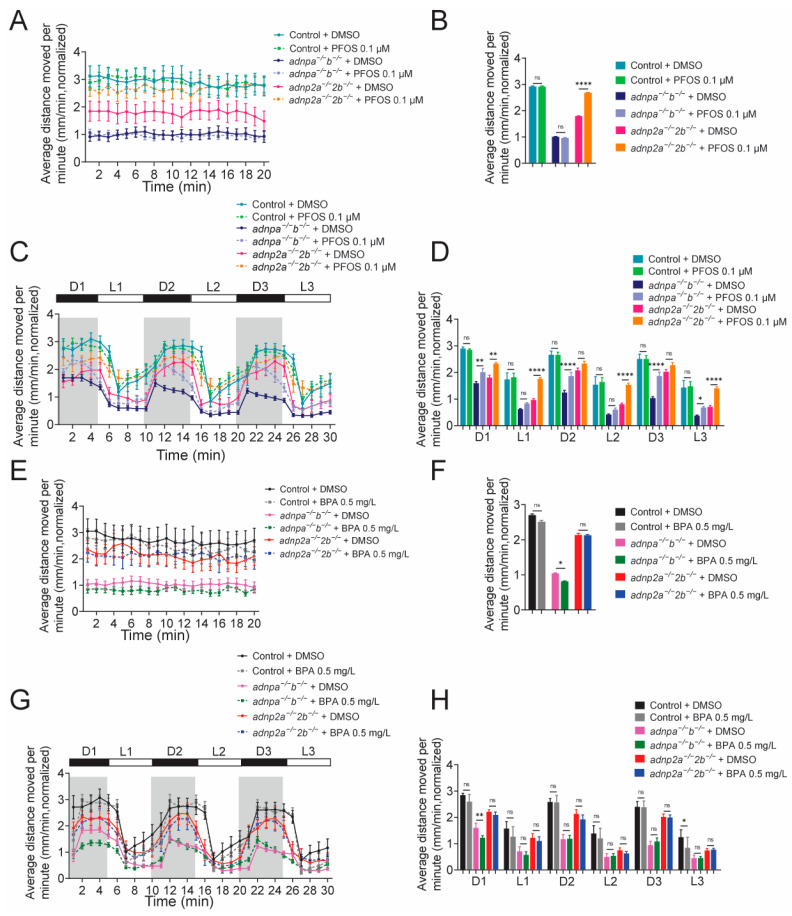
PFOS and BPA aggregate abnormal behaviors of *adnp*/*adnp2* mutants. (**A**) Average distance moved per minute of the indicated larvae with and without PFOS, under constant illumination conditions. (**B**) Quantification of panel A (n = 30 per group). (**C**) Average distance moved per minute of the indicated larvae with and without PFOS, under light/dark shift condition. (**D**) Quantification of panel (**C**) (n = 30 per group). (**E**) Average distance moved per minute of the indicated larvae with and without BPA, under light/dark shift conditions. (**F**) Quantification of panel (**E**) (n = 30 per group). (**G**) Average distance moved per minute of the indicated larvae with and without BPA under light/dark shift conditions. (**H**) Quantification of panel (**G**) (n = 30 per group). Data are presented as mean ± SEM; * *p* < 0.05, ** *p* < 0.01, **** *p* < 0.0001, ns represents no significance.

**Table 1 ijms-25-09469-t001:** Fifteen-day sequencing result of *adnp2b^+/^^−^* in-cross offspring.

Genotype	*adnp2b^+/+^*	*adnp2b^+/^* * ^−^ *	*adnp2b* * ^−/^ * * ^−^ *	Total
Number	56	62	0	118

## Data Availability

Data are contained within the article.

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
