# Peer review of "Genetic and Environmental Factors Co-Contributing to Behavioral Abnormalities in adnp/adnp2 Mutant Zebrafish"

_ijms, 2024, doi:10.3390/ijms25179469_

Round 1
Reviewer 1 Report
Comments and Suggestions for Authors
In this study, Wang et al. generated zebrafish models and used multiple behavior tests to investigate the role of adnp/adnp2 mutations on brain development and neurological defects. Using the RNA-seq, qPCR and whole-mount ISH, they showed expressions of a serial of pathway/molecules were changed in mutant brain tissues. At last, they treated the mutant fishes with EEDs, and found these environmental reagents could aggravate “abnormal” behaviors. In generally, all the experiments are well-designed and elegantly executed. Although mechanisms are weak, the body of manuscript is composed of huge amount of experimental data/information, which could be appreciated by fish research communities and neuroscientists.
Below are several points which need to be considered during the revision:
1, Introduction part, the author emphasizes lots on adnp/adnp2 with ASD. However, the mutant fishes have microcephaly phenotype (adnpa-/-b-/-; adnp2a-/-2b-/-). Thus, the reviewer suggested to properly discuss this phenotype/mechanism with data from the RNA-seq data.
2. page 1, line 36-37, NDD should include microcephaly.
3, figure 3G+H: if the reviewer understands correctly, control fishes have tendency to stay in light zone (from the heatmap).
4. figure 3I, there may be mislabeling of “adnp2-/-b-/-“ (it should be adnpa-/-b-/-).
5. Behavior tests revealed that adnp2a-/-2b+/- and adnpa-/-b-/- fishes have distinct behavior. And adnp2a-/-2b+/- fishes behaves more like controls (fig3h, j; figure 4). Can the author comments (reason) on this?
6. It would be ideal to show the HE staining on brain sections from the control and mutants’ embryos or adult fishes.
Author Response
Dear Reviewer;
First and foremost, I would like to extend my sincere gratitude for your thorough review and valuable feedback on our manuscript. Your comments are both forward-looking and comprehensive, providing us with invaluable guidance. In response to your suggestions, we have made detailed revisions to enhance the quality of our paper. We hope that these changes meet your expectations. Should there be any further issues, please do not hesitate to let us know.
Thank you once again for dedicating your time and effort to improving the quality of our work. We have highlighted all the revised sections in yellow in the manuscript for your convenience.
Comment 1: Introduction part, the author emphasizes lots on adnp/adnp2 with ASD. However, the mutant fishes have microcephaly phenotype (adnpa-/-b-/-; adnp2a-/-2b-/-). Thus, the reviewer suggested to properly discuss this phenotype/mechanism with data from the RNA-seq data.
Response 1: Thank you for pointing this out. In response to your suggestion, we have elaborated on the main mechanisms by which the loss of adnp and adnp2 in zebrafish may lead to cerebellar hypoplasia or atrophy and behavioral changes, as detailed on page 15, lines 461-483 of the discussion section. We believe this revision enhances the completeness, comprehensiveness, and logical coherence of the manuscript. The specific additions are as follows:
Overall, the downregulated KEGG and GO results from RNA-seq indicate that zebrafish lacking adnp exhibit a reduction in several critical pathways involved in cellular metabolism, protein synthesis, neural development, apoptosis, and immune response [10, 39, 55, 56]. The observed decrease in cerebellar size and behavioral abnormalities in these mutant zebrafish can be attributed to disruptions in these pathways, which impact neuron proliferation, differentiation, and function, leading to cerebellar underdevelopment or atrophy. Specifically, the affected pathways include those related to cell proliferation (e.g., MAPK and Ribosome pathways), neural development (e.g., GABAergic and Glutamatergic synapse pathways), metabolic regulation (e.g., Retinol metabolism and Oxidative phosphorylation pathways), and immune and inflammatory responses (e.g., Antigen processing and IL-17 signaling pathways). Conversely, zebrafish lacking adnp2 primarily influence development and differentiation pathways, which directly impact cerebellar development and cell differentiation processes (e.g., Estrogen signaling pathway, ECM-Receptor interaction, Steroid biosynthesis, Focal adhesion, and PI3K-Akt signaling pathway) [57], as well as metabolic regulation pathways (e.g., Protein digestion and absorption, Biotin metabolism, Glycine, Serine, and Threonine metabolism, and Thermogenesis). Additionally, pathways related to cellular function and health are also affected (including Oxidative phosphorylation, Apoptosis, Parkinson's disease, Huntington's disease and Non-Alcoholic fatty liver disease). These disruptions impact cellular function, health, and stress responses, further influencing cerebellar development. In summary, transcriptomic data suggest that the absence of adnp and adnp2 in zebrafish affects neuron proliferation, differentiation, and function, which may be the primary mechanisms leading to cerebellar underdevelopment, atrophy, and behavioral changes.
The following references have been added to pages 23, lines 820-821:
- Hedges, V. L.; Ebner, T. J.; Meisel, R. L.; Mermelstein, P. G., The cerebellum as a target for estrogen action. Front Neuroendocrinol 2012, 33, (4), 403-11.
Comment 2: page 1, line 36-37, NDD should include microcephaly.
Response 2: Thank you very much for your valuable feedback. It is indeed necessary to include "microcephaly" in this context. Based on your suggestion, we have updated the sentence on page 1, lines 35-37 from "including autism spectrum disorder (ASD), attention deficit/hyperactivity disorder (ADHD), intellectual disability (ID), and schizophrenia (SZ) " to "including autism spectrum disorder (ASD), attention deficit/hyperactivity disorder (ADHD), intellectual disability (ID), schizophrenia (SZ) and microcephaly."
Comment 3: figure 3G+H: if the reviewer understands correctly, control fishes have tendency to stay in light zone (from the heatmap).
Response 3: We greatly appreciate you pointing out this issue. The original images did indeed have some clarity problems. Since there are differences in the heatmaps presented by each fish under each genotype, I personally feel that the old image shows adnp2 as already tending more towards the bright area compared to the control group. Based on your suggestion, to make Figure 3G clearer and avoid confusion, could we replace the original image with a new one that aligns with the statistical results?
The original images are shown below:
The revised images are shown below:
Comment 4: figure 3I, there may be mislabeling of "adnp2-/-b-/-" (it should be adnpa-/-b-/-).
Response 4: Thank you for pointing out this issue. We apologize for our oversight and appreciate you bringing this to our attention. We have corrected "adnp2-/-b-/-" to "adnpa-/-b-/-" in Figure 3I of the revised manuscript as follows:
Comment 5: Behavior tests revealed that adnp2a-/-2b+/- and adnpa-/-b-/- fishes have distinct behavior. And adnp2a-/-2b+/- fishes behaves more like controls (fig3h, j; figure 4). Can the author comments (reason) on this?
Response 5: Thank you very much for addressing this issue. In this study, we used classic ASD behavioral testing methods to demonstrate that adnpa-/-; adnpb-/- adult zebrafish exhibited clear ASD-like characteristics, including stereotypical and repetitive behaviors, impaired social abilities, anxiety-like behaviors, and abnormal aggression. Meanwhile, adnp2a-/-; adnp2b+/- also displayed similar or different behavioral changes, though the extent of these changes is less pronounced compared to adnpa-/-; adnpb-/-. We believe this may be due to the following factors:
- Behavioral Changes Related to adnp2 Dosage: In juvenile behavioral assays, we found that the extent of behavioral changes in adnp2 mutants is related to the dosage of adnp2b. This suggests that adnp2a and adnp2b genes may have a synergistic effect in regulating behavioral changes. In adult fish experiments, the heterozygous condition of adnp2b in the adnp2a knockout background might have reduced the severity of behavioral abnormalities.
- Limitations of Behavioral Tests: The behavioral tests used in our study were designed based on the high correlation between adnp and ASD. We first verified whether zebrafish lacking adnp exhibit ASD-like behavioral characteristics and then assessed whether the loss of adnp2 produces similar or different behavioral changes compared to adnp mutants. Therefore, we employed classic ASD behavioral tests, including stereotypical and repetitive behaviors, social abilities, anxiety-like behaviors, and aggression. However, these tests might be insufficient to measure behavioral abnormalities related to adnp2, such as delusions in schizophrenia or recurrent traumatic memories and nightmares in post-traumatic stress.
These two factors may be significant reasons why behavioral changes in adnp2 mutants are less pronounced than those in adnp mutants. We also observed this issue in our experiments and thus employed multiple approaches to assess behavioral changes, such as using both the novel tank and three-tank tests for social behavior, and combining the light-dark box and novel tank tests for anxiety-like behaviors. Through these comprehensive and detailed investigations, we found that adnp2 also exhibits similar behavioral changes to adnp, including social abnormalities and anxiety-like behaviors.
Comment 6: It would be ideal to show the HE staining on brain sections from the control and mutants’ embryos or adult fishes.
Response 6: Thank you very much for your suggestion. We realize that our previous consideration was not comprehensive. Based on your feedback, we performed HE staining on brain tissue from 5-month-old male fish of different genotypes at the same location, which further confirms the significant differences in brain structure in zebrafish lacking adnp. The corresponding results have been added to Supplementary Figure 3D. However, to obtain results quickly, we initially used frozen sections for HE staining, which resulted in somewhat lower image quality. We are also conducting HE staining on paraffin sections, and will provide higher-quality HE images subsequently. Is this acceptable?
Additionally, we have added the following to the main text, page 4, lines 132-133: "The HE staining results showed that the brain structure of adnpa-/-; adnpb-/- fish had also undergone significant changes (Supplementary Figure S1D). We have also added the following legend information to Supplementary Figure S1D: “(D) HE staining of adult fish brain tissue. Male, 5 months. Scale bar=500 μm.” The revised Supplementary Figure S1 is shown below:
Supplementary figure 1. Related to Figure 1. (A) Death curve of adnp2b deficient (adnp2b deficient indicated adnp2b+/- in-cross) offspring within 16 days. (B) Histogram of angle between Meckel’s cartilage, angle between ceratohyal bone, length of palatoquadrate. (C) Relative cerebellar area of control, adnpa-/-; adnpb-/- and adnp2a-/-; adnp2b+/- adults at 4 months old. (D) HE staining of adult fish brain tissue. Male, 5 months. Scale bar=500 μm. Data are presented as mean± SEM; *P < 0.05, ****P < 0.0001. ns represents no significance.
Thank you once again for your valuable feedback and time. We believe these revisions will significantly enhance the quality of the manuscript and hope you find our responses satisfactory. If you have any further questions or suggestions, please feel free to let us know.
Best regards,
Yuhua Sun
Institute of Hydrobiology, Chinese Academy of Sciences
University of Chinese Academy of Sciences
No. 7 Donghu South Road. Wuhan. Hubei Province. PR China. 430072
Tel: +86-27-68780718
E-mail: [email protected]

Reviewer 2 Report
Comments and Suggestions for Authors
Comments to the Authors
Line 41. This cyte is focus on the role of ADNP2 gene with PTSD (post-traumatic stress disorder) [9-12]. It its better to indicate a reference about the roe of ADNP2 gene in neurodevelopment and autism.
Since ADNP is a vasoactive intestinal peptide (VIP) response gene, which regulates autophagy by inhibiting P53 expression, I was wondering if there is some kind of gastrointestinal problems together behavioral deficits in these mutants. Please, indicate if exist intestinal alterations associated with these mutants.
In general, the introduction should be improved with a fluent content since its seems a telephram with very short sentences. Please, improve the content with a more fluent english style and content that can relationship several aspects of the introduction.
Line 78. In zebrafish genome, the adnp gene has two copies, adnpa and adnpb [8]. adnpa-/- ; adnpb- 78 /- line has been established and maintained in our lab. Additionally, zebrafish adnp2 also 79 has two paralogues: adnp2a and adnp2b [17]. Please, describe with more detail how these mutants were created in your lab. How adnp2a and adnp2b double mutants. adnp2a-/- and 92 adnp2b+/- are generate in adult zebrafish? Explain it.
Line 116. Explain the use of alcian blue staining. Is this marker identify that means this marker and decribe the meaning of length of palatoquadrate in your study.
How the open field test could be readapted for the evaluation of locomotor and thigmotactic activities in these zebrafish mutants? The open field is more appropiate for behavioral studies in rodent models.
Line 136. Zebrafish show natural thigmotaxis to adapt to new environments. Shall you explain the meaning of this sentence? Control zebrafish larvae tent to swim at the periphery zone. Froma molecular view point, explain the reason by whihc both adnpa-/- ; adnpb-/- and adnp2 mutant larvae spent more time in the center zone than controls (Figure 2D; Supplementary Figure S2C, D).
Line 146. Colectivelly, the disruption of adnp and adnp2 gens alters the locomotor activity, thigmotaxis, and the response to light/dark shifts of zebrafish larvae. Please, shall you explain the molecular pathways that can explain these behavioral disturbances in these mutants?
The figure indicate n=68 in the foot of figure. I don,t understand this size sample. Please, shall you clarify the size sample for seach mutant in your study?
The stress and anxiety-like behaviors was evaluated by the light-dark box test. Shall you explain the idoneity of this anxiety test in these zefrafish mutants?
Line 206. How the social interaction was investigated in these mutants?
Liine 212. Please, explain the three-tank test in this model.ç
You conclude that these mutants have features of ASD, including stereotypical and repetitive behaviors, impaired social interaction, anxiety-like behavior, abnormal aggression, whereas adnp2a-/- ; adnp2b +/- fish exhibited anxiety and social preference problem.
Shal you explain these behavioral alterations with published findings in humans with autisms or PSDT síndrome (for example). Have been these mutations associated to the progression of autism of anxiety-related behaviors.?
Line 178. The figure 3E is difficult to see [(E) Representative swimming trajectories showing the stereotypical and repetitive behaviors of adnpa-/- ; adnpb-/- adults)].
Line 234. To investigate the molecular mechanism by which disruption of adnp and adnp2 leads 234 to abnormal behaviors, bulk RNA sequencing was performed for brain tissues isolated from 7 dpf larvae. What does mean bulk RNA sequeuncig?
Line 242. Please, indicate the exactly altered genes of GABAergic synapse, circadian rhythm, apoptosis, cholinergic or glutamatergic synapses.
What does mean the noto donwregulation in your study?
ELine 267. Explain who apoptosis (bcl-2, caspase3, caspase9), and microtubule dynamics genes are affected in thse mutants (mapta, 267 map6a)?
In general, the discussion should include more information about some non described genes as for example Noto or BDNF although the table shows many genes and some of them are misssing in the discussion and are really play a important role in neurodevelopmental diseases.
The genes related to autophagy such as atg5 are significantly upregulated in adnpa-/- ; adnpb-/- larval brains, indicating potential compensatory mechanisms or dysregulation in cellular homeostasis. Shall you explain these compensatory mechanisms?
Explain the contribution of p53 and BDNF and Huc in your mutants? How glutamate transporters contribute to these observed behavioral déficits in your mutants (see figure-7)?
Line 309. The levels of HuC and α-Tubulin proteins in adnp and adnp2 mutant larvae were significantly decreased. Shall you explain how HuC total fluorescence intensity has been normalized in figure 6C,D
Line 356. The figure 7 D includes asterisc * outside the bar of study groups. Please, revise figure 7 (D) quantification of panel C (n=30 per group).
Line 400. ADNP/NAP are known to regulate Tau or other microtubule end-binding proteins (Maps) by binding to microtubule end-binding protein 3 (EB3) to jointly ensure neuronal survival and function [references 16, 41-44]. So, it is possible that Tau hyperphosphorilation could induce dehavioral deficits in these zebrafish mutants? How does contribute Atg5 to autophagy in these mutants?
Explain the connextion between DNA methylation, and environmental pollutants as posible inductors of epigenome changes in these mutants
The conclusion shows the involvement of some affected genes in behavioral déficits in these mutants. However, the information about BDNF and Noto genes (for example) is null in the discussion. Please, add this information.
Please, also include future perspectives and limitations of your study.
The discussion should include more relevant information about some non describe genes, which are evalauted in figure 5C.
The methodology is elegant and all experiments were well designed with adecuate controls.
The discussion should be improved following my advice (see my comments to Authors) and the conclussion must be include some of these relevant altered genes.
My Decision is Accept with minnor revision
Thanks¡
Author Response
Dear Reviewer;
Thank you for taking the time to review our manuscript and provide valuable feedback. We greatly appreciate your detailed comments and suggestions. We have carefully addressed each of your points, and your insights have been extremely helpful to our research. Based on your feedback, we believe that the rigor, logic, and quality of the manuscript have been significantly improved. Below are our detailed responses, and we hope they adequately address your concerns.
Thank you for your valuable time and suggestions. We have highlighted all the revised sections in yellow in the manuscript for your convenience.
Comment 1: Line 41. This cyte is focus on the role of ADNP2 gene with PTSD (post-traumatic stress disorder) [9-12]. It is better to indicate a reference about the roe of ADNP2 gene in neurodevelopment and autism.
Response 1: Thank you very much for your feedback on this matter. This suggestion will help provide a more detailed introduction to the function of ADNP2. According to your recommendation, we have revised the sentence from: "ADNP is notably associated with ASD, whereas the ADNP2 gene is closely linked with SZ/PTSD (post-traumatic stress disorder) [9-12]" to: "ADNP is notably associated with ASD and developmental delay (DD), while the ADNP2 gene is closely linked with SZ, post-traumatic stress disorder (PTSD), ASD, and DD [9-14].", on page 1 lines 39-41.
We have also inserted the following references on page 20, line 713-716:
- He, X.; Zhao, P.; Huang, Y.; Cai, X.; Bi, B.; Lin, J., [Genome-wide copy number microarray analysis for a boy with autism]. Zhonghua Yi Xue Yi Chuan Xue Za Zhi 2019, 36, (2), 157-160.
- Lu, J.; Zhu, Y.; Williams, S.; Watts, M.; Tonta, M. A.; Coleman, H. A.; Parkington, H. C.; Claudianos, C., Autism-associated miR-873 regulates ARID1B, SHANK3 and NRXN2 involved in neurodevelopment. Transl Psychiatry 2020, 10, (1), 418.
Comment 2: Since ADNP is a vasoactive intestinal peptide (VIP) response gene, which regulates autophagy by inhibiting P53 expression, I was wondering if there is some kind of gastrointestinal problems together behavioral deficits in these mutants. Please, indicate if exist intestinal alterations associated with these mutants.
Response 2: Thank you very much for your attention to this issue. In our study, we primarily focused on the impact of adnp deficiency on the brain's neural system, and thus did not conduct additional analyses on the effects of adnp on the gut. Based on your feedback, to explore whether adnp deficiency leads to changes in gut structure, we performed HE staining on cross-sections of the intestines from 5-month-old male fish of different genotypes. Preliminary results show that there are no significant changes in the gut structure of adnp mutants, including the 1: intestinal villi, 2: basement membrane, and 3: goblet cells. To respond to your query promptly, we initially used frozen sections for HE staining, which may have resulted in less complete cross-sections and lower-quality images. To obtain clearer results, we also conducted HE staining on paraffin-embedded sections, which requires additional time. We will provide higher-quality images once the paraffin sections are completed. Does this arrangement work for you?
Below are the preliminary HE staining results:
Figure legend: HE staining results of the cross-sectional view of the intestinal tract in male adult fish, aged 5 months. The lower image is an enlarged view of the black rectangular area in the upper image. 1: intestinal villi, 2: basement membrane, 3: goblet cells.
Comment 3: In general, the introduction should be improved with a fluent content since its seems a telephram with very short sentences. Please, improve the content with a more fluent english style and content that can relationship several aspects of the introduction.
Response 3: Thank you very much for your suggestions on improving the flow and coherence of the introduction. A smoother and more coherent text will indeed help readers follow and understand the paper better. Therefore, we have revised the introduction to enhance its flow and coherence. The specific changes are as follows:
on page 2 lines 47-58. “ADNP and ADNP2 both have zinc finger and homeobox domain, suggesting that they function as transcription factors [19]. Mouse models of ASD that mimic ADNP mutations have been established, and have facilitated the understanding of the etiology of the diseases [20, 21]. Zebrafish have been widely used for modeling NDDs [22-24]. However, adnp/adnp2 mutant zebrafish models that recapitulate core features of NDDs have not been reported.” revised to “ADNP and ADNP2 both have zinc finger and homeobox domain, suggesting that they may perform similar or identical functions within cells [19]. Our laboratory's previous research showed that ADNP influences neuronal differentiation by regulating the Wnt/β-catenin signaling pathway[8], but the specific role of ADNP2 in the nervous system remains unclear. Although mouse models of ASD mimicking Adnp mutations have been established and have helped elucidate the etiology of related diseases, the role of ADNP in the nervous system and how to treat diseases caused by ADNP deletion are still not fully understood [20, 21]. Moreover, there are no reports on the development of ADNP2 knockout lines. Zebrafish have been widely used for modeling NDDs [22-24]. In this study, we aim to comprehensively understand and investigate the intrinsic regulatory mechanisms of ADNP and ADNP2 on nervous system function by constructing adnp/adnp2 mutant zebrafish models.”
Comment 4: Line 78. In zebrafish genome, the adnp gene has two copies, adnpa and adnpb [8]. adnpa-/- ; adnpb- 78 /- line has been established and maintained in our lab. Additionally, zebrafish adnp2 also 79 has two paralogues: adnp2a and adnp2b [17]. Please, describe with more detail how these mutants were created in your lab. How adnp2a and adnp2b double mutants. adnp2a-/- and 92 adnp2b+/- are generate in adult zebrafish? Explain it.
Response 4: Thank you for your suggestion to include detailed information on the construction of mutants. In response, we have added a reference to our previous work on constructing adnp mutants on page 2 line 83-84 “adnpa-/-; adnpb-/- line has been established and maintained in our lab [8].” which provides a detailed description of how the mutants were constructed.
Additionally, to avoid overwhelming the results section with extensive details on mutant construction, we have provided a comprehensive description of the adnp2a-/-2b+/- construction process in the Methods section of the manuscript, on page 16 lines 516-541. Does this approach meet your expectations? If you have any further concerns or suggestions, we will be happy to make additional adjustments based on your guidance.
Comment 5: Line 116. Explain the use of alcian blue staining. Is this marker identify that means this marker and decribe the meaning of length of palatoquadrate in your study.
Response 5: Thank you for your feedback on the Alcian blue staining. Following your suggestion, we have refined the explanation of the markers of Alcian blue staining: “Moreover, adnpa-/-; adnpb-/- embryos displayed craniofacial deficits, which was confirmed by alcian blue staining”, which now reads: “Moreover, the craniofacial deficits observed in the adnpa-/-; adnpb-/- larvae were confirmed through alcian blue staining, a technique that binds to sulfated glycosaminoglycans present in the cartilage matrix and thereby indicates the development of cartilage in zebrafish”, on page 4 lines 120-123.
Furthermore, the length of the palatoquadrate can be used to describe or measure the size and shape of the palatoquadrate, as it develops into the upper jaw. In the text, this measurement, along with the angle between Meckel’s cartilage and the ceratohyal bone structures, is used to indicate the development of craniofacial cartilage.
Comment 6: How the open field test could be readapted for the evaluation of locomotor and thigmotactic activities in these zebrafish mutants? The open field is more appropiate for behavioral studies in rodent models.
Response 6: Thank you for raising the question about the open field test. This is indeed one of the key factors supporting the feasibility and rationale for using zebrafish in behavioral studies in this paper. The open field test is a behavioral assay used to assess locomotion, thigmotaxis, and anxiety states in animals. Originally designed for rodents such as mice and rats, it has recently been adapted for behavioral studies in zebrafish. Zebrafish exhibit valuable behavioral attributes, including locomotor and thigmotactic activities. The open field test provides valuable insights into their locomotion, spatial exploration capabilities, and responses to environmental stimuli, making it particularly significant in genetics and pharmacology research. Overall, while the application of the open field test in zebrafish and rodents has its nuances, the fundamental principles are similar, aiming to evaluate animal movement, exploration, and anxiety levels. With appropriate adjustments and optimizations, this test can yield valuable behavioral data across different animal models. I hope this explanation addresses your concerns.
Comment 7: Line 136: Zebrafish exhibit natural thigmotaxis in new environments. Could you explain the meaning of this sentence? Control zebrafish larvae tend to swim at the periphery. From a molecular perspective, explain why both adnpa-/-; adnpb-/- and adnp2 mutant larvae spend more time in the center zone compared to controls (Figure 2D; Supplementary Figures S2C, D).
Response 7: Thank you very much for your detailed questions regarding the thigmotaxis test. Thigmotaxis in zebrafish refers to a tactile orientation behavior where zebrafish tend to swim close to the edges or surfaces of objects. The statement "Control zebrafish larvae tend to swim at the periphery" means that "the time spent in the peripheral zone by mutant zebrafish is significantly reduced compared to the wild-type controls." Therefore, based on your feedback, to clarify our statement, we have revised the sentence on page 4, lines 144-145 from "Control zebrafish larvae tend to swim at the periphery zone" to "Both adnpa-/-; adnpb-/- and adnp2 mutant larvae spend significantly less time in the peripheral zone compared to the wild-type controls."
From a molecular perspective, our understanding of why both adnpa-/-; adnpb-/- and adnp2 mutant larvae spend more time in the center zone compared to controls is: Zebrafish lacking adnp exhibit downregulation of several critical pathways related to cellular metabolism, protein synthesis, neural development, apoptosis, and immune response, as indicated by the downregulated KEGG and GO results from RNA-seq. On the other hand, zebrafish deficient in adnp2 primarily affect development and differentiation pathways, which directly impact cerebellar development and cell differentiation processes. Transcriptomic data suggest that the absence of adnp and adnp2 in zebrafish affects neuron proliferation, differentiation, and function, which may be the key reason why adnp and adnp2 mutant larvae spend more time in the central zone compared to the control group.
Comment 8: Line 146: Collectively, the disruption of adnp and adnp2 genes alters locomotor activity, thigmotaxis, and the response to light/dark shifts in zebrafish larvae. Please, shall you explain the molecular pathways that can explain these behavioral disturbances in these mutants?
Response 8: Thank you for focusing on the molecular mechanisms affecting behavioral changes such as locomotor activity, thigmotaxis, and responses to light/dark shifts. Your attention greatly enhances the quality of the manuscript. After considering your and other reviewers' feedback on the molecular mechanisms underlying the behavioral changes in mutants, we believe it is crucial to provide a comprehensive and detailed discussion of the potential mechanisms behind the behavioral abnormalities caused by the loss of the adnp family. This will help improve the overall completeness, structure, and coherence of the manuscript. Specifically, we have added the following section from lines 461-483 on pages 15 of the discussion:
Overall, the downregulated KEGG and GO results from RNA-seq indicate that zebrafish lacking adnp exhibit a reduction in several critical pathways involved in cellular metabolism, protein synthesis, neural development, apoptosis, and immune response [10, 39, 55, 56]. The observed decrease in cerebellar size and behavioral abnormalities in these mutant zebrafish can be attributed to disruptions in these pathways, which impact neuron proliferation, differentiation, and function, leading to cerebellar underdevelopment or atrophy. Specifically, the affected pathways include those related to cell proliferation (e.g., MAPK and Ribosome pathways), neural development (e.g., GABAergic and Glutamatergic synapse pathways), metabolic regulation (e.g., Retinol metabolism and Oxidative phosphorylation pathways), and immune and inflammatory responses (e.g., Antigen processing and IL-17 signaling pathways). Conversely, zebrafish lacking adnp2 primarily influence development and differentiation pathways, which directly impact cerebellar development and cell differentiation processes (e.g., Estrogen signaling pathway, ECM-Receptor interaction, Steroid biosynthesis, Focal adhesion, and PI3K-Akt signaling pathway) [57], as well as metabolic regulation pathways (e.g., Protein digestion and absorption, Biotin metabolism, Glycine, Serine, and Threonine metabolism, and Thermogenesis). Additionally, pathways related to cellular function and health are also affected (including Oxidative phosphorylation, Apoptosis, Parkinson's disease, Huntington's disease and Non-Alcoholic fatty liver disease). These disruptions impact cellular function, health, and stress responses, further influencing cerebellar development. In summary, transcriptomic data suggest that the absence of adnp and adnp2 in zebrafish affects neuron proliferation, differentiation, and function, which may be the primary mechanisms leading to cerebellar underdevelopment, atrophy, and behavioral changes.
The following references have been added to pages 23, lines 820-821:
- Hedges, V. L.; Ebner, T. J.; Meisel, R. L.; Mermelstein, P. G., The cerebellum as a target for estrogen action. Front Neuroendocrinol 2012, 33, (4), 403-11.
Comment 9: The figure indicate n=68 in the foot of figure. I don’t understand this size sample. Please, shall you clarify the size sample for search mutant in your study?
Response 9: Thank you for focusing on this detail. The "n=68 per group" indicated in Figure 2A means that in the juvenile fish open field test, there were 68 samples for each genotype, including wild type group, adnpa-/-b-/- group, adnp2a-/-2b+/+ group, adnp2a-/-2b+/- group, and adnp2a-/-2b-/- group, making a total of five groups.
Comment 10: The stress and anxiety-like behaviors were evaluated by the light-dark box test. Shall you explain the idoneity of this anxiety test in these zefrafish mutants?
Response 10: Thank you for raising this question. The light-dark box test is a commonly used behavioral method to assess anxiety-like behaviors in animals. This test involves observing zebrafish behavior in an environment that includes both light and dark areas. Typically, zebrafish prefer to stay in the dark areas to minimize exposure, and anxiety-like behavior is indicated by reduced time spent in the light area. Our study found that adnp mutants spent significantly less time in and entered the light area less frequently compared to the control group, indicating pronounced anxiety-like behavior. Thus, the light-dark box test effectively detects stress and anxiety-like behaviors in adnp mutants. Based on your suggestion, we have added information regarding the applicability of the light-dark box test in these zebrafish mutants on page 7, lines 208-210. The added content is as follows:
It indicated that the adnpa-/-b-/- shows altered natural responses to the light-dark environment compared to the controls, effectively assessing the anxiety behavior of the adnpa-/-b-/-.
Comment 11: Line 206: How was social interaction investigated in these mutants?
Response 11: Thank you for raising this question. To assess changes in social interactions of mutant zebrafish, we used the zebrafish shoaling test. This experiment aims to evaluate zebrafish behavior patterns and sociality in a social environment. In the experiment, we placed five adult male zebrafish from each group into tanks and allowed them to swim freely without interference. We recorded metrics such as swimming patterns, shoal size, shoal stability, and distances between individuals. Specifically, we assessed the social abilities of the mutants by measuring the average inter-fish distance in a group and the overall average distance. The results showed that, compared to the control group, adnpa-/-b-/- and adnp2a-/-2b+/- exhibited significant changes in average inter-fish distance and average distance, indicating impaired social abilities in the mutants.
Comment 12: Line 212: Please explain the three-tank test in this model.
Response 12: Thank you for raising this question. In the three-tank test, the tank dimensions are 40 cm × 15 cm × 15 cm, divided into three areas: a central area measuring 20 cm × 15 cm × 15 cm, and two side areas each measuring 10 cm × 15 cm × 15 cm. The right-side area contains 5 zebrafish of the same species, while the left-side area serves as a control without fish. The central area is subdivided into three sections: “empty zone,” “middle zone,” and “social zone.” We quantify social behavior by calculating the time zebrafish spend in these areas. Results show that adnpa-/-b-/- and adnp2a-/-2b+/- spend significantly less time in the “social zone” and more time in the “empty zone” and “middle zone,” indicating reduced contact with conspecifics and impaired social behavior.
Based on your suggestion, we have revised the sentence on page 7, lines 224-234 from “Both adnpa-/-; adnpb-/- and adnp2a-/-; adnp2b+/- fish spent less time at ‘social zone’ than the controls, and spent more time at the ‘middle- and empty zones’.” to: “The three-tank test was divided into three compartments. No zebrafish were placed in the left compartments, while five conspecific zebrafish were placed in the right compartment. The central testing compartment was also divided into three zones, namely ‘empty zone,’ ‘middle zone,’ and ‘social zone.’ By comparing the time spent by different genotypes of zebrafish in each zone with the time spent interacting with conspecifics in the right compartment, we assessed changes in social behavior. The results showed that both adnpa-/-; adnpb-/- and adnp2a-/-; adnp2b+/- fish spent significantly less time in the ‘social zone’ compared to controls and spent more time in the ‘middle zone’ and ‘empty zone.’ This indicated that the social behavior of both adnpa-/-; adnpb-/- and adnp2a-/-; adnp2b+/- fish were abnormal compared to the controls.”
Comment 13: You conclude that these mutants have features of ASD, including stereotypical and repetitive behaviors, impaired social interaction, anxiety-like behavior, abnormal aggression, whereas adnp2a-/-; adnp2b+/- fish exhibited anxiety and social preference problem .Shal you explain these behavioral alterations with published findings in humans with autisms or PSDT síndrome (for example). Have been these mutations associated to the progression of autism of anxiety-related behaviors.?
Response 13: Thank you for raising this question. Symptoms of ASD in humans typically include impaired social interaction and repetitive behaviors, often accompanied by comorbid conditions such as stress, anxiety, and aggression. Given the strong association between ADNP and ASD, we used six classic adult zebrafish behavioral assays for ASD in our experiments. This approach helps to identify typical ASD-related behavioral changes in adnp mutants and reveals behavioral alterations in adnp2 mutants.
PTSD (post-traumatic stress disorder) can be assessed by evaluating impaired social function and anxiety-related behaviors. Since ADNP2 is linked to PTSD, we also observed relevant behavioral changes. Establishing adnp and adnp2 mutant lines not only provides models for drug screening for neurodevelopmental disorders (including ASD) but also lays a theoretical foundation for studying the mechanisms of neurodevelopmental disorders caused by ADNP or ADNP2 deficiencies, including ASD and related anxiety behaviors.
Comment 14: Line 178.The figure 3E is difficult to see [(E) Representative swimming trajectories showing the stereotypical and repetitive behaviors of adnpa-/-; adnpb-/- adults)].
Response 14: Thank you for pointing this out. When we were compiling data on stereotypic and repetitive behaviors, we spent considerable time referencing literature on similar behaviors and sought guidance from experts to accurately analyze our data. In Figure 3E, we have demonstrated that adnpa-/-b-/- exhibit representative stereotypic and repetitive behaviors, including: small circling (rotating or moving in a small circular area), walling (repeatedly swimming along the tank walls), cornering (staying or repeatedly performing activities in corners), and "figure 8" (displaying repetitive movements in an 8-shaped pattern). Below, we list two references on zebrafish stereotypic and repetitive behavior statistics, including representative trajectory diagrams of these behaviors. If you have any additional comments or suggestions, please let us know, and we will make further revisions based on your feedback. Does this arrangement work for you?
Zheng J, Long F, Cao X, Xiong B, Li Y. Knockout of Katnal2 Leads to Autism-like Behaviors and Developmental Delay in Zebrafish. Int J Mol Sci. 2022 Jul 29;23(15):8389. doi: 10.3390/ijms23158389. PMID: 35955524; PMCID: PMC9368773.
Liu CX, Li CY, Hu CC, Wang Y, Lin J, Jiang YH, Li Q, Xu X. CRISPR/Cas9-induced shank3b mutant zebrafish display autism-like behaviors. Mol Autism. 2018 Apr 2;9:23. doi: 10.1186/s13229-018-0204-x. PMID: 29619162; PMCID: PMC5879542.
Comment 15: Line 234: To investigate the molecular mechanisms behind the abnormal behaviors caused by adnp and adnp2 disruption, bulk RNA sequencing was performed on brain tissues from 7 dpf larvae. What does "bulk RNA sequencing" entail?
Response 15: "Bulk RNA sequencing" refers to extracting total RNA from a sample, encompassing all types of RNA present. In this study, we collected head tissue from 7-day-old juvenile fish and removed the non-brain tissue from the lower jaw to enrich for brain tissue cells as much as possible. After extracting total RNA from the isolated tissue, we performed RNA sequencing to analyze the overall transcriptome of the tissue.
Comment 16: Line 242: Please specify the exact genes affected in GABAergic synapses, circadian rhythm, apoptosis, cholinergic or glutamatergic synapses.
Response 16: Thank you for your suggestion, which has enriched and improved our transcriptome data. Based on your feedback, we have added the exact genes for the specified signaling pathways on page 9, lines 261-265. The revised sentence is as follows:
down-regulated DEGs showed enriched terms such as GABAergic synapse (including gabrr2b, gabra6a), circadian rhythm (including cry1aa, per2, nr1d1), apoptosis (including atf4a, atf4b), cholinergic synapse (including gng3, gngt2a), and glutamatergic synapse (including vesicular glutamate transporters (VGluT) related genes: slc38a2, slc1a8a).
Comment 17: What does mean the noto donwregulation in your study?
Response 17: Thank you for raising this point. The dysfunction of the noto gene may be associated with various neurodevelopmental disorders, including brain development abnormalities, cognitive impairments, and motor coordination issues. Research indicates that mutations in the noto gene in model organisms such as zebrafish and mice can lead to defects in nervous system development, affecting behavior and neurological function. In this study, "noto downregulation" refers to the significant reduction in noto gene expression observed in the brain tissue RNA-seq results from adnp knockout samples.
Comment 18: Line 267. Explain who apoptosis (bcl-2, caspase3, caspase9), and microtubule dynamics genes are affected in thse mutants (mapta, 267 map6a)?
Response 18: Thank you for your feedback. It may not have been clearly expressed; we intended to indicate that in adnp mutants, the apoptosis-related genes (such as bcl-2, caspase-3, caspase-9) and microtubule-associated protein genes (such as mapta, map6a) are all significantly downregulated.
Comment 19: In general, the discussion should include more information about some non described genes as for example Noto or BDNF although the table shows many genes and some of them are missing in the discussion and are really play an important role in neurodevelopmental diseases.
Response 19: Thank you very much for pointing this out. We may have assumed that genes such as noto and bdnf are well-known and thus overlooked providing detailed information about their importance. Based on your feedback, we have added the relevant information to lines 450-460 on page 15 of the discussion section, revising “We also found that early response genes, and neurodegenerative and neuropsychiatry genes, were down-regulated in adnp mutant larvae. In fact, previous studies have shown that early response was abnormal in NDDs.” to “We also found that early response genes, such as egr2/4, fosab, and jun, which are rapidly and transiently expressed in response to cellular stimuli, play crucial roles in neurodevelopment, memory formation, and stress responses [51, 52]. These genes are downregulated in adnp mutant larvae. Similarly, genes associated with neurodegenerative and neuropsychiatric disorders, such as bdnf, chd2, es1, ngf, and noto, are also downregulated in adnp mutants. For instance, bdnf (brain-derived neurotrophic factor) is involved in the growth and development of glutamatergic and GABAergic synapses and regulates dopaminergic neurotransmission. Abnormal expression of bdnf is linked to major diseases such as Huntington's disease, Alzheimer's disease, schizophrenia, and anxiety disorders. noto plays a crucial role in early embryonic development, affecting the formation and function of the nervous system [53, 54].”.
And accordingly, references have been added to lines 808-815 on pages 22-23, as follows::
- Swanberg, S. E.; Nagarajan, R. P.; Peddada, S.; Yasui, D. H.; LaSalle, J. M., Reciprocal co-regulation of EGR2 and MECP2 is disrupted in Rett syndrome and autism. Hum Mol Genet 2009, 18, (3), 525-34.
- Davis, S.; Bozon, B.; Laroche, S., How necessary is the activation of the immediate early gene zif268 in synaptic plasticity and learning? Behav Brain Res 2003, 142, (1-2), 17-30.
- Colucci-D'Amato, L.; Speranza, L.; Volpicelli, F., Neurotrophic Factor BDNF, Physiological Functions and Therapeutic Potential in Depression, Neurodegeneration and Brain Cancer. Int J Mol Sci 2020, 21, (20).
- Andreska, T.; Aufmkolk, S.; Sauer, M.; Blum, R., High abundance of BDNF within glutamatergic presynapses of cultured hippocampal neurons. Front Cell Neurosci 2014, 8, 107.
Comment 20: The genes related to autophagy such as atg5 are significantly upregulated in adnpa-/-; adnpb-/- larval brains, indicating potential compensatory mechanisms or dysregulation in cellular homeostasis. Shall you explain these compensatory mechanisms?
Response 20: Thank you very much for your detailed attention to the issue of "potential compensatory mechanisms or dysregulation in cellular homeostasis." Our results indicate that adnp deficiency in zebrafish leads to a significant increase in autophagy-related genes (such as p53 and atg5) and a significant decrease in apoptosis-related genes (such as bcl-2, caspase-3, and caspase-9). This suggests that the relationship between autophagy and apoptosis is complex and dynamic, possibly reflecting a compensatory adjustment by zebrafish to survive or avoid irreversible damage to the nervous system after adnp loss.
Another possibility is that ADNP, as a neuroprotective peptide, provides protection to nerve cells. Previous studies have shown that ADNP can protect cells by inhibiting the expression of P53. Therefore, when adnp is absent, the homeostasis between autophagy and apoptosis may be disrupted, leading to an inability of cells to effectively maintain balance under stress.
Comment 21: Explain the contribution of p53 and BDNF and Huc in your mutants? How glutamate transporters contribute to these observed behavioral déficits in your mutants (see figure-7)?
Response 21: Thank you for pointing this out. In Figure 7, we show that environmental pollutants exacerbate behavioral abnormalities in adnpa-/-b-/- and adnpa-/-b-/- larvae. We acknowledge that our study lacks a detailed mechanistic analysis in this regard, which is a limitation. However, we believe that in adnpa-/-b-/-, the upregulation of p53 indicates abnormal autophagy regulation due to adnp loss. The downregulation of bdnf suggests disruptions in neuro-related pathways, while the downregulation of the pan-neuronal marker HuC indicates overall abnormalities in the nervous system in adnpa-/-b-/- and adnp2a-/-2b-/- larvae. The balance between excitatory and inhibitory neurons is crucial for neural homeostasis, and abnormal expression of glutamate transporters disrupts this balance, leading to behavioral abnormalities. Exposure to environmental pollutants further exacerbates these behavioral changes.
Comment 22: Line 309. The levels of HuC and α-Tubulin proteins in adnp and adnp2 mutant larvae were significantly decreased. Shall you explain how HuC total fluorescence intensity has been normalized in figure 6C, D
Response 22: We appreciate your attention to this detail. The method we used to normalize the fluorescence intensity of HuC and α-Tubulin proteins involved dividing the fluorescence intensity measurements from different groups (including the wild-type group) by the average fluorescence intensity value of the wild-type control group, and then performing statistical analysis.
Comment 23: Line 356. The figure 7 D includes asterisc * outside the bar of study groups. Please, revise figure 7 (D) quantification of panel C (n=30 per group).
Response 23: Thank you for pointing this out. Based on your feedback, we have adjusted the size of the significance markers (asterisks) and "ns" labels in Figure 7 to better fit each panel. The revised Figure 7 is as follows:
Comment 24: Line 400. ADNP/NAP are known to regulate Tau or other microtubule end-binding proteins (Maps) by binding to microtubule end-binding protein 3 (EB3) to jointly ensure neuronal survival and function [references 16, 41-44]. So, it is possible that Tau hyperphosphorylation could induce behavioral deficits in these zebrafish mutants? How does contribute Atg5 to autophagy in these mutants?
Response 24: Thank you for addressing this issue. We believe that Tau protein hyperphosphorylation could be a contributing factor to the behavioral deficits observed in zebrafish mutants. Tau protein hyperphosphorylation is a characteristic feature in many neurodegenerative diseases, such as Alzheimer’s disease, cognitive disorders, and other Tau-related diseases. The NAP motif in ADNP interacts directly with microtubule end-binding protein 3 (EB3) to regulate interactions between microtubules, Tau, and EBs, thereby maintaining the normal structure and function of the neuronal fiber network. Hyperphosphorylation of Tau may lead to decreased microtubule stability in nerve cells, affecting neuronal function and synaptic transmission. This could result in behavioral abnormalities in zebrafish mutants, such as impaired motor coordination and learning and memory deficits. Therefore, we propose that adnp loss might lead to behavioral defects in zebrafish mutants through Tau hyperphosphorylation.
The atg5 gene encodes a protein that is a crucial regulator in the autophagy process, involved in the formation of autophagosomes and essential for cellular homeostasis. Consequently, we suggest that in zebrafish lacking adnp, an abnormal autophagy-apoptosis balance, possibly due to increased p53 and atg5, might be formed, thereby affecting cell survival.
Comment 25: Explain the connection between DNA methylation, and environmental pollutants as possible inductors of epigenome changes in these mutants
Response 25: We appreciate your attention to this issue. The subsequent toxicological experiments in this study aim to explore how genetic and environmental factors jointly contribute to neurodevelopmental disorders. ADNP and ADNP2 are involved in chromatin remodeling processes, particularly in regulating the SWI/SNF complex, including mechanisms such as DNA methylation and histone modification. Environmental factors may influence these neurodevelopmentally relevant genes by altering DNA methylation and histone modifications, thereby modulating gene expression. Such regulation could potentially lead to the onset or exacerbation of neurodevelopmental disorders.
Comment 26: The conclusion shows the involvement of some affected genes in behavioral déficits in these mutants. However, the information about BDNF and Noto genes (for example) is null in the discussion. Please, add this information.
Response 26: Thank you for pointing this out. Based on your feedback, we have added detailed information about these genes in the discussion section, lines 450 to 460 on page 15. Please refer to the response to comment 19 for the details.
Comment 27: Please, also, include future perspectives and limitations of your study.
Response 27: Thank you for your attention to this detail. Adding this section enhances the completeness and depth of the article. Based on your suggestion, we have added a "Future Directions and Limitations" section on lines 646-657 of page 19. The content is as follows:
- Future Directions and Limitations
This study aims to investigate the molecular mechanisms of neurodevelopmental disorders associated with the loss of adnp family genes in zebrafish, focusing primarily on their functions in brain neurons. However, the absence of a head-specific conditional knockout may introduce additional effects from the loss of adnp family genes in other body regions. Additionally, our study did not delve into the single-cell level effects of adnp and adnp2 loss on brain neural system function, limiting the depth of our findings.
We have developed a model of adnp family neurodevelopmental disorders in zebrafish and examined the molecular changes following the loss of adnp family genes. We hope that this model will be useful in future treatments for adnp family deficiencies or ASD and will provide a platform for large-scale drug screening for neurodevelopmental diseases caused by adnp family gene loss.
Comment 28: The discussion should include more relevant information about some non describe genes, which are evalauted in figure 5C.
Response 28: Thank you for your continued attention to the need for incorporating information about changes in the results section. Based on your feedback and the comments on the related issues, we have added the following information to the discussion, specifically on page 14, lines 430 to 434:
A set of steroidogenic and neuroendocrine genes, including the growth and development-regulating gene fosab, the estrogen synthesis gene cyp191a1, the neuropeptide gene galn, the corticosteroid gene hsd11b2, the steroidogenesis gene star, the insulin-like 3 peptide gene insl3, and the progesterone receptor gene pgr, were down-regulated in adnpa-/-; adnpb-/- larvae.
Comment 29: The discussion should be improved following my advice (see my comments to Authors) and the conclusion must be include some of these relevant altered genes.
Response 29: Thank you for your detailed feedback on the discussion section. We have revised it accordingly, as detailed in comments 19, 26, and 28.
Additionally, as per your request, we have updated the conclusion to reflect the changes in the relevant genes. The conclusion has been modified from:“Neurons, autophagy, apoptosis, circadian clock and estrogen signaling pathways might mediate adnp/adnp2 in the development of neural disorders.” to “The neuronal markers involved include vglut1, vglut2a, vglut2b, gad1b, gad2, th, bdnf, and noto; autophagy-related genes include p53, atg5, and atg3; apoptosis-related genes include bcl-2 and caspase3; circadian clock genes include cry1, cry2, per1, per2, and nr1d1; immediate early response genes include egr2, egr4, and jun; and estrogen signaling pathways include igf3, prl2, galn, cyp19a1a, hsd11b2, star, insl3, and pgr. These factors may mediate the development of neural disorders in adnp/adnp2.”, on page 14, lines 430 to 434:
Thank you once again for your valuable feedback and time. We believe these changes will significantly enhance the quality of the manuscript. If you have any further questions or suggestions, please do not hesitate to let us know.
Best regards,
Yuhua Sun
Institute of Hydrobiology, Chinese Academy of Sciences
University of Chinese Academy of Sciences
No. 7 Donghu South Road. Wuhan. Hubei Province. PR China. 430072
Tel: +86-27-68780718
E-mail: [email protected]

Reviewer 3 Report
Comments and Suggestions for Authors
The manuscript entitled “Genetic and Environmental Factors co-Contributing to Behavioral Abnormalities in adnp/adnp2 Mutant Zebrafish”, by Y. Wang et al., is an important contribution to the knowledge of mutations in ADNP and ADNP2 which in human are associated with developmental nerve abnormalities such as schizophrenia and autism spectrum disorders (ASD). In this article, zebrafish was used as a model and CRISPR/Cas9 gene editing technology to generate dnp and dnp2 mutants which exhibited developmental delays, brain deficits, and core behavioral features of NDDs. Results demonstrate altered gene expression of genes related genes to synaptic transmission, autophagy, apoptosis, microtubule dynamics, hormone signaling, and circadian rhythm regulation. Data were validated throughout whole mount in situ hybridization (WISH) and real-time quantitative PCR (qRT-PCR). Additional experiments demonstrate that exposure to environmental endocrine disruptors (EEDs) aggravated behavioral abnormalities in adnp and adnp2 mutants.
In this reviewer's opinion, the work contains new and important information and, above all, offers an excellent model for preclinical studies. The work is well-structured and easy to follow. The material and methods are very comprehensive and the techniques are described in sufficient detail to allow replication. The results are clear and well presented. It is worth noting the high quality and information offered by figures 5D and 6A.
Author Response
Dear Reviewer,
Thank you for your detailed review of our manuscript titled "Genetic and Environmental Factors co-Contributing to Behavioral Abnormalities in adnp/adnp2 Mutant Zebrafish." We greatly appreciate your positive feedback and recognition of our research into the role of ADNP and ADNP2 mutations in developmental neural disorders.
We are pleased that you view our study as a significant contribution to the field and an excellent preclinical model. We are also glad that you found the manuscript well-structured, easy to understand, and that the materials and methods were comprehensive and detailed enough to support replication.
We appreciate your comments on the high quality and informativeness of figures 5D and 6A. We will ensure that these figures and the entire manuscript maintain their high quality and clarity in the final version.
If you have any additional aspects or further suggestions that need clarification, please let us know. We will do our best to make any necessary revisions to enhance the quality of the manuscript.
Thank you once again for your valuable feedback and review.
Sincerely,
Yuhua Sun
